# Isotopic composition of convective rainfall in the inland tropics of Brazil

Vinicius dos Santos[1], Didier Gastmans[1], Ana María Durán-Quesada[2], Ricardo Sánchez-Murillo[3], Kazimierz Rozanski[4], Oliver Kracht[5] and Demilson de Assis Quintão[6].

[1]São Paulo State University (UNESP), Environmental Studies Center. Av. 24A, 1515, Bela Vista, 13.506-900, Rio Claro, São Paulo, Brazil. vinicius.santos16@unesp.br; didier.gastmans@unesp.br

[2]Escuela de Física & Centro de Investigación en Contaminación Ambiental & Centro de Investigaciones Geofísicas, Universidad de Costa Rica, San José 11501, Costa Rica. ana.duranquesada@ucr.ac.cr

[3]University of Texas at Arlington, Department of Earth and Environmental Sciences, 500 Yates Street, Arlington, Texas 76019, USA. ricardo.sanchezmurrillo@uta.edu

[4]Faculty of Physics and Applied Computer Science, AGH University of Krakow, al. Mickiewicza 30, 30-059 Krakow, Poland. rozanski@agh.edu.pl

[5]International Atomic Energy Agency, Isotope Hydrology Section, Vienna International Centre, P. O. Box 100, 1400 Vienna, Austria. O.Kracht@iaea.org

[6]São Paulo State University (UNESP), IPMet/Science College, Est. Mun. José Sandrin IPMET, S/N, 17.048-699, Bauru, São Paulo, Brazil. demilson.quintao@unesp.br

*Correspondence to*: Didier Gastmans (didier.gastmans@unesp.br)

**Abstract.** The tropical central-southern region of Brazil is characterized by strong convective systems. These systems provide abundant water for agro-industrial activities but also pose flood risks to large cities. Here, we present high-frequency (2-10 min) rainfall isotopic compositions (n=90 samples) to reveal the regional and local atmospheric processes controlling the isotopic variability of convective systems from 2019-2021. Isotope parameters from individual events, including initial ($\delta_{initial}$), median ($\delta_{med}$), and the difference between lowest and highest isotope values ($\Delta\delta$), and detailed meteorological data, were used in inter-event and intra-event analysis. The lower $\delta_{initial}$ values were associated with higher rainfall along Hysplit trajectories from the Amazon forest during the summer, compared to autumn and spring, when Hysplit trajectories from the Atlantic Ocean and South Brazil had lower amounts of rainfall. Consequently, there were high $\delta_{initial}$ values. This regional $\delta$-signature was conserved during certain convective intra-events, with similar values between the $\delta_{initial}$ and $\delta_{median}$. However, for other intra-events, the $\delta_{initial}$ values were altered by local processes connected to cloud features, rainfall vertical structure, and humidity conditions, resulting in increased isotopic variations ($\Delta\delta$) during intra-events. Our findings establish a novel framework for evaluating the meteorological controls on the isotopic variability of convective precipitation in tropical South America, fill the gap of high-frequency studies in this region, and generate a comprehensive meteorological dataset for future modeling studies.

## 1 Introduction

The tropical central-southern region of Brazil (CSB) is the primary contributor to the country's economy, with agriculture and agroindustry as the main sectors (Zilli et al., 2017). These economic activities are highly dependent on seasonal rainfall for irrigation and hydropower supply (Luiz Silva et al., 2019). Projected changes in the frequency of heavy and extreme rainfall events in future climate scenarios (Marengo et al., 2020; Donat et al., 2013; IPCC, 2021; Marengo et al., 2021) pose a significant threat to regional economic enterprises and power generation. Similarly, according to Marengo et al. (2021), simulations with the pre-CMIP6 models suggest that the intensification of heavy rainfall events could exacerbate the prevalence of floods and landslides in susceptible regions. Such occurrences have resulted in a total cost of US$41.7 billion over the past half-century (Marengo et al., 2020; World Meteorological Organization, 2021).

Extreme precipitation events are linked to convective systems (CS). The CS significantly contribute proportion of annual rainfall and account for a significant portion of extreme rainfall (Roca and Fiolleau, 2020). Across the tropics, diurnal surface heating amplifies convection, generating short-lived events that can occur in consecutive days. Rapid upward movement of air results in quick condensation and formation of precipitation with substantial droplets and heavy rainfall (Breugem et al., 2020; Kastman et al., 2017; Lima et al., 2010; Machado et al., 1998). This is identified by vigorous vertical development in the form of *cumulus-nimbus* and *cumulus congestus* (convective clouds) and low-level divergence (stratiform clouds) (Siqueira et al., 2005; Machado and Rossow, 1993; Zilli et al., 2017; Houze, 1989, 2004). Precipitation associated with these systems are commonly referred as convective and stratiform rainfall, and account for 45% and 46% of the total rainfall in South America, respectively (Romatschke and Houze, 2013).

Whether rainfall is convective or stratiform rainfall has been suggested to determine variations in stable isotope composition of precipitation across the tropics (Zwart et al., 2018; Sánchez-Murillo et al., 2019; Sun et al., 2019; Han et al., 2021; Aggarwal et al., 2016; Munksgaard et al., 2019).

Processes driving the variations in the isotopic composition in convective systems are more complex and less understood compared to the case of other precipitation producing systems (de Vries et al., 2022). Studies using the isotopic composition of rain and water vapor have quantified and modelled physical processes related to convection (Bony et al., 2008; Kurita, 2013). Previous studies have suggested that the isotopic composition of convective systems is connected to the integrated history of convective activity (Risi et al., 2008; Moerman et al., 2013), depth of organized convection and aggregation (Lawrence et al., 2004; Lekshmy et al., 2014; Lacour et al., 2018; Galewsky et al., 2023), microphysical processes within clouds (Aggarwal et al., 2016; Lawrence et al., 2004; Zwart et al., 2018), and cold pool dynamics (Torri, 2021). These interpretations simplified and lumped the effects of multiple rainfall timescales (e.g. monthly, daily and high frequency), providing different perspectives on convective processes, such as the regional (synoptic forcings) and local factors (e. g. microphysical processes occurring both within and below the cloud) (Kurita et al., 2009; Muller et al., 2015).

High-frequency rainfall sampling and analyses of stable isotope ratios has been used to better understand the evolution of large weather systems such as tropical cyclones and typhoons (Sun et al., 2022; Sánchez-Murillo et al., 2019; Han et al., 2021),

squall lines (Taupin et al., 1997; Risi et al., 2010; Tremoy et al., 2014) and local evaporation effects (Graf et al., 2019; Aemisegger et al., 2015; Lee and Fung, 2008). High-resolution isotope information can provide a better insight into the development of weather systems and cloud dynamics, both responsible for changes in the rain type, intensity, and inherent isotope variability during the life cycle of rainfall events (Coplen et al., 2008; Muller et al., 2015; Celle-Jeanton et al., 2004).

In this study, we used high-frequency rainfall sampling to investigate regional (moisture origin/transport, regional atmospheric circulation) and local (below-cloud processes, vertical structure of rainfall, cloud top temperature) processes that controlling the isotopic composition of convective rainfall. High-frequency rainfall was integrated with ground-based observational data (Micro Rain Radar and automatic weather station), satellite imagery (GOES-16), ERA-5 reanalysis products, and HYSPLIT trajectories to better characterize convective rainfall over the inland tropics of Brazil.

## 2 Data and Methods

### 2.1 Sampling site and weather systems

The rainfall sampling site was localized in Rio Claro city, São Paulo State (Fig. 1a). The station (-22.39°S, -47.54°W, 670 m a.s.l.) is part of the Global Network of Isotopes in Precipitation network (GNIP) and is influenced by weather systems responsible for rainfall variations and seasonality linked to the regional atmospheric circulations across the CSB region. The rainfall seasonality over CSB is associated with: (i) frontal systems (FS), represented mainly by cold fronts from southern South America acting throughout the year, and (ii) the activity of the South Atlantic Convergence Zone (SACZ) during austral summer (December to March) (Kodama, 1992; Garreaud, 2000) (Fig. 1b). These features are mostly responsible for CS development (Romatschke and Houze, 2013; Siqueira et al., 2005; Machado and Rossow, 1993) (Fig. 1c), and were captured during their passage over the Rio Claro station.

### 2.2 Rainfall sampling and isotope analyses

High-frequency rainfall sampling was conducted using a passive collector (2 to10 minutes intervals) from September 2019 to February 2021, except for April, July, and August (during winter 2020), when no rainfall was observed in the study area. The pandemic Covid-19 disrupted access to the university, thereby reducing the number of rainfall events sampled during the spring of 2020, particularly at night (e.g., lockdowns). In this study, the rainfall samples collected do not consist of consecutive day-night pairs during the same day. In total, 90 samples representing eight convective events (3 night-time and 5 day-time events) were collected. Samples were transferred to the laboratory and stored in 20 mL HDPE bottles at 4 ºC. In parallel to high-frequency sampling, monthly cumulative rainfall samples were also collected at the Rio Claro site during the study period as a contribution to the GNIP network, using the methodology recommended by the International Atomic Energy Agency (IAEA, 2014).

Rainfall samples were analyzed for stable isotope composition using Off-Axis Integrated Cavity Output Spectroscopy (Los
Gatos Research Inc.) at the Hydrogeology and Hydrochemistry laboratory of UNESP's Department of Applied Geology and
at the Chemistry School of the National University (UNA, Heredia, Costa Rica). All results are expressed in per mil relative
to Vienna Standard Mean Ocean Water (V-SMOW). The certified calibration standards used in UNESP were USGS-45 ($\delta^2$H
= −10.3 ‰, $\delta^{18}$O = −2.24 ‰), USGS-46 ($\delta^2$H = −236.0 ‰, $\delta^{18}$O = −29.80 ‰), including one internal standard (Cachoeira de
Emas - CE – $\delta^2$H = −36.1 ‰, $\delta^{18}$O = −5.36 ‰). USGS standards were used to calibrate the results on the V-SMOW2-SLAP2
scale, whereas CE was used for memory and drift corrections. At UNA, the certified standards MTW ($\delta^2$H = −130.3 ‰, $\delta^{18}$O
= −16.7 ‰), USGS45 ($\delta^2$H = −10.3 ‰, $\delta^{18}$O = −2.2 ‰), and CAS ($\delta^2$H = −64.3 ‰, $\delta^{18}$O = −8.3 ‰) were used to correct the
measurement results for memory and drift effects and to calibrate them on the V-SMOW2-SLAP2 scale (García-Santos et al.,
2022). The analytical uncertainty (1σ) was 1.2 ‰ for $\delta^2$H and 0.2 ‰ for $\delta^{18}$O for UNESP analysis and 0.38 ‰ for $\delta^2$H and
0.07 ‰ for $\delta^{18}$O for UNA analysis. Deuterium excess (*d*-excess) was calculated as: *d*-excess = $\delta^2$H - 8*$\delta^{18}$O (Dansgaard,
1964), with uncertainties (1σ) of 1.33 and 0.43 ‰, respectively. This secondary isotope parameter was used to interpret the
influence of moisture origin/transport (Sánchez-Murillo et al., 2017; Froehlich et al., 2002) and local processes (Aemisegger
et al., 2015; Muller et al., 2015; Celle-Jeanton et al., 2004).
**2.3 Meteorological data**
Automatic Weather Station (AWS) Decagon Em50 (METER) was installed near the Micro Rain Radar (MRR) (METEK) at
670 m.a.s.l, in immediate vicinity of the rainfall collection site. Meteorological data were recorded at 1 min intervals for rain
rate (RR, mm min$^{-1}$), air temperature (T, °C) and relative humidity (RH, %). The MRR data for reflectivity (Zc, dBZ), and fall
velocity (w, m s$^{-1}$) were also recorded at 1 min intervals. MRR parameters correspond to the mean values measured at the
elevation between 150 and 350 meters above surface. MRR operated at a frequency of 24.230 GHz, modulation of 0.5 – 15
MHz according to the height resolution mode. For this work, different height resolutions (31 range bin) were tested, 150 m,
200 m, 300 m and 350 m, resulting in vertical profiles of 4650 m, 6200 m, 930 0m and 10.850 m, respectively (Endries et al.,
2018). The MRR data used in the following discussion are the near-surface data (first measurement from 150 m to 350 m).
Lifting Condensation Level (LCL, meters) was computed from AWS RH and T, using expression proposed by Soderberg et
al. (2013) and rainfall amount (R, mm) was calculated during the sampling interval. GOES-16 imagery was used to identify
the convective nuclei of the cloud-top (10.35-$\mu$m, Band-13) and brightness temperature (BT, °C), at 10 min intervals during
the sampling period (Ribeiro et al., 2019; Schmit et al., 2017). The 10.35-$\mu$m BT is often used to estimate the convective cloud
depth, since the lower BT is linked to deeper cloud tops (Adler and Fenn, 1979; Roberts and Rutledge, 2003; Adler and Mack,
1986; Ribeiro et al., 2019; Machado et al., 1998). The weather systems (fronts, instabilities, and low pressure) were defined
according to the synoptic chart and meteorological technical bulletin of the Center for Weather Forecast and Climatic Studies
of the National Institute of Space Research (CPTEC/INPE) that used information of numerical models, automatic weather
stations, satellite and radar images, reanalysis data and regional atmospheric models, such as the Brazilian Global Atmospheric
Model and ETA model.

**2.4 Hysplit modeling and Reanalysis data**

The origin of air masses and moisture transport to the Rio Claro site were evaluated using the HYSPLIT (Hybrid-Single
Particle Langragian Integrated Trajectory) modeling framework (Stein et al., 2015; Soderberg et al., 2013). The trajectories of
the air masses were estimated for 240 hours prior to rainfall onset, considering the estimated time of residence of the water
vapor (Gimeno et al., 2010, 2020; van der Ent and Tuinenburg, 2017). Start time of trajectories was the same as the start time
of rainfall events. The trajectories were computed using NOAA′s meteorological data (global data assimilation system, GDAS:
1 degree, global, 2006-present), with ending elevations of the trajectories at 1500 m above the surface, taking into account the
climatological height of the Low Level Jet, within 1000–2000 m (Marengo et al., 2004). Ten-day trajectories representing
convective events were calculated as trajectory ensembles, each consisting of twenty-seven ensemble members released at
start hour of convective rainfall sample collection. Ensembles were produced by varying the initial trajectory wind speeds and
pressures, according to the HYSPLIT ensemble algorithm, in order to account for the uncertainties involved in the simulation
of individual backward trajectories (Jeelani et al., 2018). A sum of the rainfall intensity (mm hr$^{-1}$) along the trajectories was
used to analyse rainout of the moist air masses according to the Jeelani et al. (2018).
Reanalysis data were used to better understand the influence of atmospheric circulation on isotopic composition of rainfall
at the study area. ERA-5 information was used to evaluate hourly vertical integrals of eastward water vapor flux (kg m$^{-1}$ s$^{-1}$)
during convective events sampled. The Global Modeling and Assimilation Office (GMAO) data (MERRA-2, 1 hour, 0.5 x
0.625 degree, V5.12.4 were used for calculations of latent heat flux (LHF). Aqua/AIRS L3 Daily Standard Physical Retrieval
(AIRS-only) data (1 degree x 1 degree V7.0, Greenbelt, MD, USA, Goddard Earth Sciences Data and Information Services
Center) (known as GES DISC) were used for average outgoing longwave radiation (OLR). OLR values below 240 W m$^{-2}$
indicate organized deep convection (Gadgil, 2003).

**2.5 Identification of convective rainfall events**

In general, identification of convective precipitation systems was based on the vertical structure of the given precipitation
system (lack of the melting layer and bright band - BB) in the radar profiles featuring high reflectivity values (Zc > 38 dBZ)
(Houze, 1993, 1997; Steiner and Smith, 1998; Rao et al., 2008; Mehta et al., 2020; Endries et al., 2018) and satellite imagery
(Vila et al., 2008; Ribeiro et al., 2019; Siqueira et al., 2005; Machado et al., 1998). Consequently, convective rainfall was
defined in this study by (i) convective cloud nuclei observed in GOES-16 imagery, (ii) no BB detected, (iii) Zc > 38 dBZ near
to the surface and (iv) rainfall intensity (AWS) of at least 10 mm h$^{-1}$ (Klaassen, 1988) (Fig. 1c,d). The convective nuclei were
identified using GOES-16 imagery, determined as a contiguous area of at least 40 pixels with BT lower than 235K ($\leq$ -38 °C)
over Rio Claro station, according to previous studies (Ribeiro et al., 2019).

## 2.6 Preliminary assessment of local processes

Below-cloud atmospheric conditions are known to be relevant and while we acknowledge that a more robust dataset is required to provide sound conclusions, a preliminary assessment of this factor is herein included.

Since the isotopic composition of near-ground water vapor during the rainfall events was not measured, the framework proposed by Graf et al. (2019) for interpreting below-cloud effects on rainfall isotopes cannot be applied here. A semi-quantitative evaluation of those effects is demonstrated for all rainfall events, despite the need for a more substantial dataset to establish firm conclusions. This analysis considers the following assumptions: (i) median values of isotope and meteorological parameters recorded for each analysed event (Table 1) will be used in the calculations, (ii) linear interpolation of air temperature and relative humidity between the cloud base level and the ground level will be adopted to estimate the relative humidity at the cloud base ($RH_{INT}$), (iii) it will be assumed that atmosphere is saturated with water vapour at the cloud base level (RH = 100 %), and (iv) the reservoir of water vapour below the cloud base level is isotopically homogeneous (Risi et al., 2019; Sarkar et al., 2023).

Isotopic evolution of raindrops falling through unsaturated humid atmosphere beneath the cloud base level will be calculated using the generally accepted conceptual framework for isotope effects accompanying evaporation of water into a humid atmosphere (Craig and Gordon, 1965; Horita et al., 2008). Isotopic evolution of an isolated water body (e.g. falling raindrop) evaporating into a humid atmosphere can be described by the following equations (Gonfiantini, 1986):

$$\delta = \left(\delta_o - \frac{A}{B}\right)F^B + \frac{A}{B} \tag{1}$$

where

$$A = \frac{h_N\delta_A + \varepsilon_{kin} + \varepsilon_{eq}/\alpha_{eq}}{1 - h_N + \varepsilon_{kin}} \tag{2}$$

and

$$B = \frac{h_N - \varepsilon_{kin} - \varepsilon_{eq}/\alpha_{eq}}{1 - h_N + \varepsilon_{kin}} \tag{3}$$

Parameter $F$ describes the remaining fraction of the evaporating mass of water (raindrop), while $\delta_A$ stands for the isotopic composition of ambient moisture. Initial and actual isotopic compositions of the evaporating water body, expressed in $\delta$ notation, are represented by $\delta_o$ and $\delta$, respectively. The variables in equations (3) and (4) are described as:

$h_N$ – relative humidity of the ambient atmosphere, normalized to the temperature of the evaporating water body;

$\alpha_{eq}$ – temperature-dependent equilibrium fractionation factor, derived from empirical equations proposed by Horita and Wesolowski (1994);

$\varepsilon_{eq}$ – equilibrium fractionation coefficient: $\varepsilon_{eq} = \alpha_{eq} - 1$ (4)

$\varepsilon_{kin}$ - kinetic fractionation coefficient; $\varepsilon_{kin} = \alpha_{kin} - 1$ (5)

The kinetic fractionation coefficient is a linear function of the relative humidity deficit in the ambient atmosphere (Gat, 2001; Horita et al., 2008):

$$\varepsilon_{kin} = n \cdot \varepsilon_{diff}(1 - h_N) \tag{6}$$
where $n$ describes a turbulence parameter, varying from zero to one and $\varepsilon_{diff}$ is the kinetic fractionation coefficient associated
with diffusion of water isotopologues in air.
The value of $n$ is controlled mainly by wind conditions prevailing over the evaporating surface. It quantifies the apparent
reduction of $\varepsilon_{diff}$ due to the impact of turbulent transport. The value of $n = 0.5$, was adopted in the calculations, following the
results of laboratory experiments with evaporation of water drops in a humid atmosphere reported by Stewart (1975).
Following this same publication, the value of the F parameter for each event was computed based on the rate of change of
evaporated drop radius as a function of ambient relative humidity (Stewart, 1975). Droplets with a drop size distribution of
1mm are assumed based on previous studies in this region of study (Zawadzki and Antonio, 1988; Cecchini et al., 2014).
Travel time of raindrops drops from the cloud base to the surface was derived from the position of LCL level and the terminal
velocity of drops. It was further assumed in the calculations that the difference between drop temperature and ambient air
temperature is small, thus allowing to use ambient humidity instead to normalized humidity. Although this assumption may
result in an over-estimation of the impact of partial evaporation of raindrops on their isotope characteristics, the effect is
expected to be small due to high ambient relative humidities (> 90 %) used in the calculations.
**2.7 Statistical tests**
The Shapiro-Wilk test was applied to verify that the data distribution was normal (parametric) or non-normal (non-parametric)
(Shapiro, S. S.; Wilk, 1965). A significant difference (p-value < 0.05) indicates a non-parametric distribution. A Spearman
rank correlation test was used for nonparametric distribution data, whereas Pearson's linear correlation test was applied for
parametric data. Correlation tests were conducted between isotopes ($\delta^{18}O$ and $d$-excess) and meteorological data (AWS and
MRR variables) during the same time interval and from individual events. Correlation tests were not applied to GOES-16 BT
and reanalysis data due to their temporal resolution, which reduced the number of samples. All tests were performed with
significance levels defined by a p-value < 0.05, using the R statistical package (R Core Team, 2023).
A statistical analysis was carried out to characterize regional and local influences, in accordance with He et al. (2018). The
initial isotope data of the events ($\delta_{initial}$) closely reflects the initial air mass or vapor from which the precipitation originates.
The $\delta_{initial}$ and median ($\delta_{med}$) values were employed to identify regional influences in inter-event analysis. Also, the difference
($\Delta\delta$) between the lowest $\delta^{18}O$ and the highest $\delta^{18}O$ value represents the local change in $\delta$-value during the intra-event (Muller
et al., 2015; He et al., 2018).
**3 Results**
**3.1  Isotopic and synoptic characteristics**

The isotopic composition of monthly rainfall exhibits clear seasonal variations between September 2019 and February 2021 (Fig. 2a). Seasonal variability was characterized by wet (low $\delta^{18}O$) and dry (high $\delta^{18}O$) seasons (austral summer and autumn-spring, respectively). High-frequency sampling of convective events could not be done uniformly during the study period, but it is still evident that median $\delta^{18}O$ values of high-frequency sampling events (black symbols in Fig. 2a) follow the seasonal isotope variability.

The summer months were characterized by the influence of convective activity, reflected in high latent heat flux and lower OLR (Fig. 2c). During autumn and spring, significant lower latent heat flux and higher OLR were associated with less convective development (Houze, 1997, 1989). The formation of convective rainfall may not be primarily controlled by diurnal thermal convection, as rainfall is more likely to be associated with frontal systems (Siqueira and Machado, 2004), as observed in the rainfall episodes during autumn and spring.

A significant influence of the cold fronts was observed before, during, and after their passage over the study area (Fig. 2a). During autumn and spring, the convective events of 2019/11/05, 2020/11/18, and 2020/05/23 were associated with cold fronts in the study area. On 2020/06/09, changes in the regional atmosphere over the state of São Paulo caused convective rainfall due to an instability (frontal) system resulting from a cold front settling over the southern region of Brazil. During the summer season, convective rainfall also occurred on 2020/02/01 and 2021/02/24 due to cold fronts and instability (frontal), respectively. In addition, the thermal convection of the continental region caused atmospheric ascent via surface heating in the inland of Brazil, leading to a system responsible for the convective rainfall event on 2020/01/30. As a result of the interaction between thermal convection and the incursion of the frontal system, a low-pressure system (frontal) was responsible for the convective rainfall event on 2020/02/10.

Table 1 presents an overview of the sampling, isotope parameters ($\delta_{initial}$, $\delta_{med}$, $\Delta\delta$) and median values of meteorological variables from individual events. Sampled events had a duration of 141 or fewer minutes. The T and Twd exhibited small differences among the events. In contrast, RR, RH, LCL, Zc, w, and BT varied considerably between events. The maximum recorded values for these parameters were 97 %, 489 m, 46 dBZ, 8 m s$^{-1}$ and -63 °C, respectively.

Isotope values varied among convective events, with a range of -11.0 ‰, -92.8 ‰ and +15.7 ‰ for median values of $\delta^{18}O$, $\delta^2H$ and $d$-excess, respectively. The maximum differences between the $\delta_{initial}$ and $\delta_{med}$ for $\delta^{18}O$, $\delta^2H$, and $d$-excess were 1.6 ‰, 9.1 ‰, and 9.5 ‰, respectively. The maximum $\Delta\delta$ values for all isotopes parameters, $\delta^{18}O$, $\delta^2H$ and $d$-excess were 7.3 ‰, 43.0 ‰ and 19.2 ‰, respectively.

**3.2. Inter-event variability of the isotope parameters**

Hysplit air mass back-trajectories revealed three main locations as moisture origin during the presence of convective rainfall: Amazon forest, Atlantic Ocean, and southern Brazil (Fig. 3). The sourcing of moisture for rainfall over Rio Claro varies

seasonally and spatially, suggesting complex interactions in moisture transport and mixing that strongly influence the initial isotopic composition of rainfall throughout the year (Table 1).

Summer rainfall events were characterized by the trajectory and length of moist air masses arriving from the Amazon forest (2020/02/10, 2020/02/01, and 2020/01/30) (Fig. 3a). As a result, there was a large amount of rainfall along Hysplit trajectories. Rainfall amounts were 177 mm, 126 mm and 78 mm, respectively, for these dates. Remarkably, these events exhibited very similar isotope characteristics ($\delta^2 H_{initial}$, $\delta^{18}O_{initial}$) (Table 1). In contrast, the event on 2021/02/24 presented higher $\delta_{initial}$ values, reflecting the oceanic moisture influence (Fig. 3a), with a lowest amount of rainfall (53 mm) along Hysplit trajectory.

Based on ERA-5, the vertically integrated eastward vapor flux corroborates the influence of a distinct mechanism for moisture transport and $\delta_{initial}$ values. Negative values for vertical vapor fluxes over the Amazon forest during sampled convective events in summer (Fig. 4a, b, d) clearly illustrate a westward moisture flux from the Atlantic Ocean to the Amazon forest. Positive values in the central-southern region of Brazil indicate moisture being transported eastward from the Amazon forest. However, these moisture fluxes were not observed on 2021/02/24 when the eastward vapor flux was positive with high values over the Atlantic Ocean (250 ~ 750 kg m$^{-1}$ s$^{-1}$).

The autumn convective events on 2020/05/23 and 2020/06/09 revealed a significant continental origin of moist air masses (from south-western Brazil). In addition, during the second event, the Amazon-type trajectory started in the southern Atlantic and did not reach the boundary of the rainforest (Fig. 3b). In both autumn events, there was the lowest amount of rainfall (4 mm) along Hysplit trajectories. On 2020/05/23 negative vertical flux values (-500 ~ -250 kg m$^{-1}$s$^{-1}$) were observed in south-western Brazil, indicating moisture transport from the Atlantic Ocean to the continent. This favored a vapor flux (500 ~ 750 kg m$^{-1}$s$^{-1}$) from western Brazil to the study area (Figure 4f). On 2020/06/09, there were slightly negative values (-250 ~ 0 kg m$^{-1}$s$^{-1}$) of eastward vapor flux in the Amazon forest, indicating less influence from rainforest moisture. Conversely, positive vapor flux values (250 ~ 500 kg m$^{-1}$s$^{-1}$) were observed in the western part of continental Brazil.

Two events in the spring season (Fig. 3c) also showed contrasting origin of moisture and initial *d*-excess values, despite only slight differences in $\delta^{18}O_{initial}$ (Table 1). The mean trajectory on 2020/11/18 clearly belongs to the Amazon category, although it only passed over the south-eastern boundary of the Amazon rainforest and had a much shorter length and lower rainfall along Hysplit trajectory (23 mm) compared to the Amazon trajectories observed during the summer season. Thus, positive values of the eastward vapor flux (250 ~ 750 kg m$^{-1}$ s$^{-1}$) were not distributed along the Amazon forest to the Atlantic Ocean as typically observed (Fig. 4h). The mean trajectory on 2019/11/05 the eastward vapor flux (> 500 kg m$^{-1}$ s$^{-1}$, Fig. 4g) were circling around Rio Claro, indicating the continental moisture origin (from southern Brazil), and low amount of rainfall along Hysplit trajectory of 8 mm.

**3.3 Intra-event variability of the isotope and meteorological parameters**

The temporal evolution of isotope characteristics and selected meteorological parameters of convective rainfall are shown in Fig. 5 (summer) and Fig. 6 (autumn-spring). The study emphasizes the lack of pattern in the measured values for reflectivity (Zc) in the vertical profile. Only higher Zc values were observed near the surface (from 2km to 200m), which indicates an increase in rain rates. Despite the similar vertical structure, the temporal evolution varied considerably among events. Furthermore, the GOES-16 BT shows unique temporal patterns among events.

The differences in $\Delta\delta$ observed between convective events were explained by intra-events (refer to Table 1) and how local factors may affect the regional isotopic signature as illustrated by the inter-event analysis.

### 3.3.1. Summer intra-events

Lower values of $\Delta\delta^{18}O$ were observed on the 2020/02/01 and 2020/01/30 compared to higher $\Delta\delta^{18}O$ values observed on the 2020/02/10 and 2021/02/24. In contrast, all summer events exhibit high $\Delta\delta$ values for $d$-excess (Table 1). Despite this variation in isotopic amplitude, the evolution of these events is characterized by different amounts of available humidity (Table 1 and Table 2). For the 2021/02/24 event, lower humidity values were observed below the cloud ($RH_{INT}$ = 93 %) and at the surface (RH 78 ~ 88 %, median value 86 %). The other events had higher humidity conditions ($RH_{INT}$ = > 96 % and RH > 90 %). Nevertheless, only 2021/02/24 and 2020/02/10 show d-excess values lower than 10 ‰, suggesting that the specific local factors can influence the variations in the isotopic composition of the precipitation, as shown below for each event.

Specifically, the events on 2020/02/01 (Fig. 5c,e) and 2020/01/30 (Fig. 5d,f) showed consistent $\delta^{18}O$ trends (-11.6 ~ -10.0 ‰ and -10.6 ~ -9.6 ‰, respectively). In contrast, these events showed an inverted V-shaped (from 11.3 ~ 15.3 ‰ to 15.4 ~ 7.0‰) and V-shaped (from 20.8 ~ 11.4 ‰ to 14.6 ~ 16.2 ‰) patterns for $d$-excess, respectively. The patterns of rainfall intensity were similar for both events, with high rainfall amount at the beginning of event, decreasing over the time. In BT values, decreased (-50 ~ -65 °C) and constant variations (-52 ~ -53 °C) occurred on 2020/02/01 and 2020/01/30 events, respectively. The strong and significant (p < 0.0001) correlations were observed between isotopic composition and MRR parameters for 2020/02/01: $\delta^{18}O$-Zc (r = -0.9), $\delta^{18}O$-w (r = -0.9), $d$-excess-Zc (r = 0.9) and $d$-excess-w (r = 0.9), while there were no correlations between isotopic composition and meteorological parameters for 2020/01/30.

On 2021/02/24 (Fig. 5i,k) and 2020/02/10 (Fig. 5j,l), notable fluctuations were observed in both the isotope and meteorological parameters. On 2021/02/24, $\delta^{18}O$ varied from -7.9 ~ -4.4 ‰, and $d$-excess varied from 1.2 to 18.4 ‰. The evolution of the event was characterized by varying local weather conditions, as evidenced by a larger BT range (-38 ~ -57 °C). Radar reflectivity is displayed in a vertical profile, illustrating these changes, with larger Zc values during the event (red colors in Fig. 5g). As a result, three peaks of maximum rainfall amount were observed, which corresponded to the distinct $\delta^{18}O$ and for $d$-excess values: at 15:49 local time (2.6 mm, -7.6‰ and 13.0 ‰), at 16:24 (3.1 mm, -6.9 ‰ and 8.4 ‰) and at 17:28 (3.3 mm, -7.9 ‰ and 17.9 ‰), respectively. Also, strong, and significant (p < 0.05) correlation was observed between $\delta^{18}O$-R (r = -0.8), $d$-excess-R (r = -0.6) and MRR parameter, $\delta^{18}O$-Zc (r = -0.6) and $d$-excess-Zc (r = -0.5).

On 2020/02/10, $\delta^{18}O$ showed a variation from -15.2 ~ -7.9 ‰ and for $d$-excess from 4.8 ~ 21.4 ‰. During the beginning
of the event and until 21:03 local time, high BT values (-16 ~ -45 °C) corresponded to the higher Zc values (red colors in Fig.
5h) and high RH (~ 97 %). After this time, lower Zc and lowest BT values were observed (-45 ~ -57 °C). There were two
breakpoints in the rainfall trend (increasing to decreasing) corresponding to the change in isotope values ($\delta^{18}O$ and $d$-excess),
occurring at 20:36 (4.8 to 3.2 mm, -13.9 to -9.5 ‰ and 15.7 to 9.4 ‰) and 21:57 (2.0 to 0.8 mm, -14.9 to -7.9 ‰ and 21.4 to
4.8‰) respectively. In addition, for this event strong and significant ($p < 0.05$) correlation was observed only between $\delta^{18}O$-
RH (r = -0.5) and $d$-excess-RH (r = 0.5).

### 3.3.2 Autumn and spring intra-events

Lower $\Delta\delta^{18}O$ values were observed during autumn and spring events in comparison to summer events. Both autumn and spring
events showed higher $\Delta\delta$ values for $d$-excess when compared to summer events. For the events on 2020/05/23 ($RH_{INT}$ = 93 %,
RH 78 ~ 89 %, with median of 87 %) and 2020/11/18 ($RH_{INT}$ = 92 % and RH 70 ~ 90 %, with median of 85 %), lower humidity
conditions were recorded, whereas for all other events, humidity conditions were high ($RH_{INT}$ = > 97 % and RH = > 90%) as
show in Tables 1 and 2.
For autumn events on 2020/06/09 (Fig. 6a,c,e) and 2020/05/23 (Fig. 6b,d,f), a slight increase trend (-3.7 ~ -1.5‰) and
stationary trend (-2.6 ~ -2.7‰) were observed regarding $\delta^{18}O$. On the other hand, for the same events, $d$-excess showed a W-
shaped trend (17.7 ~ 6.3 ‰, during the last part of the event) and V-shaped pattern (16.7 ~ 19.0 ‰), respectively. Both events
demonstrated a decrease in rainfall amount: from 6.2 to 0.2 mm on 2020/06/09, 2020, and from 2.6 to 0.2 mm on 2020/05/23.
Additionally, the range of BT increased from -55°C to -35 °C and from -60°C to -52 °C, respectively. Strong and significant
($p < 0.05$) correlations were observed between isotopic and surface meteorological parameters during the event on 2020/06/09,
$\delta^{18}O$-RH (r = 0.5), $\delta^{18}O$-T (r = -0.6), $d$-excess-RH (r = -0.6), and $d$-excess-T (r = 0.7). However, no significant correlations
were found during the event on 2020/05/23.
Spring convective events exhibited contrasting variations in isotopes and meteorological conditions. On 2019/11/05 (Fig.
6g,i,k), slight fluctuations were observed in $\delta^{18}O$ (-3.0 ~ -1.7 ‰, slightly increasing-trend), while $d$-excess values were higher
(21.0 ~ 28.0 ‰, decreasing trend). This slight fluctuations in $\delta^{18}O$ values correspond to the constant and higher Zc near surface.
This is evidenced by the highest and significant ($p < 0.0003$) correlations observed between isotopic and MRR parameters,
$\delta^{18}O$-Zc (r = -0.7), $\delta^{18}O$-w (r = -0.7), and $d$-excess-w (r = 0.6). In contrast, these fluctuations were not related with changes in
rainfall amount (0.6 ~ 5.0 mm) and BT (-65 ~ -62 °C).
On 2020/11/18, two distinct steps revealed a decreasing trend in $\delta^{18}O$ (-2.7 ~ -5.4 ‰), and a substantial increasing trend in
$d$-excess (10.2 ~ 23.1 ‰) (Fig. 6h,j,l). Between 15:10 and 16:05 local time, the vertical profile of the MRR exhibited variable
Zc values, with concomitant decreases in both BT values (-62 and -65 °C) and $\delta^{18}O$ (-2.7 ~ -4.0 ‰) and increase in both rainfall
(1.2 ~ 2.0 mm), $d$-excess (10.2 ~ 19.6 ‰) and RH (70 ~ 82 %) values. After this period, Zc values increased closer to the
surface, resulting in a slight decrease in temperature (-65 ~ -63 °C). Additionally, $\delta^{18}O$, $d$-excess, rainfall amount and RH
fluctuated (-3.8 ~ -5.4 ‰, 18.0 ~ 23.1 ‰, 1.8 ~ 2.2 mm and 84 ~ 90 %, respectively). Regardless of this, no significant
correlations were found due to the considerable variations between isotopic and rainfall, as well as BT and MRR parameters.
The significant ($p < 0.001$) correlations were only observed for $\delta^{18}O$-RH ($r = -0.9$), $\delta^{18}O$-T ($r = 0.9$), $d$-excess-RH ($r = 0.9$),
and $d$-excess-T ($r = -0.9$).

## 4. Discussion

Detailed evaluations of isotopic variability in convective rainfall were provided by both inter- and intra-events. Such separation
between inter- and intra-events allows for improved evaluation of fractionation processes that occurred during moisture
transport towards the formation of local rainfall. Generally, during the summer, thermal conditions dominate convective
processes, while during autumn and spring, convective rainfall is associated with frontal systems (Fig. 2). It is crucial to
quantify these synoptic variations to understand seasonal differences in atmospheric conditions, which affect moisture source
and transport across seasons. Thus, the $\delta_{initial}$ values are influenced by vapor origin, convective activity, and weather systems,
which may be further modified by local processes, resulting in distinct values of $\delta_{med}$ and large $\Delta\delta$.
The key regional and local controls of the isotopic composition of convective rainfall are, respectively: (i) rainfall of moist
air masses during their transport in the atmosphere, from the source region(s) to the collection site showed by inter-event
analysis, and (ii) local effects associated with convective cloud characteristics, vertical rainfall structure and near-surface
humidity conditions.

### 4.1 Regional atmospheric controls

Regional aspects of atmospheric moisture transport to the Rio Claro site were illustrated in HYSPLIT backward trajectories
(Fig. 3) and maps of vertically integrated moisture flux in the region (Fig. 4). Most of moist air masses arriving at Rio Claro
during summer (2020/02/10, 2020/02/01, and 2020/01/30) exhibited a common origin in the equatorial Atlantic Ocean and
were subjected to a long rainfall of moist air masses, extending over several thousand kilometers. Along this pathway, air
masses interacted with the Amazon forest. Intensive recycling of moisture leads to a small continental gradient of $\delta$-values of
rainfall across the Amazon forest (Salati et al., 1979; Rozanski et al., 1993) and elevated $d$-excess (Gat, J. R., & Matsui, 1991).
At Rio Claro, the arriving air masses are depleted in heavy isotopes ($\delta_{initial} \leq -10.0$ ‰) due to enhanced amount of rainfall along
the trajectories ($\geq 78$ mm), after the southeastern deflection from the Andes, with consistent initial $d$-excess higher than +10.0
‰, inherited through the interaction of maritime moisture with the Amazon forest. In contrast, the summer event on 2021/02/24
was influenced by oceanic moisture and had a short trajectory compared to the other summer events, as indicated by the lower

amount of rainfall along the Hysplit trajectory (53 mm), which explains the higher $\delta_{initial}$ values ($\delta^{18}O$ = -7.6 ‰ and $d$-excess = +13 ‰).

The convective events representing spring and autumn season exhibited substantially shorter trajectories suggesting that the atmospheric "pump" transporting moisture from the equatorial Atlantic Ocean to the Amazon forest was much weaker or non-existent during this time of the year. As a result, those trajectories were characterized by a reduction in the amount of rainfall along the trajectories and enriched $\delta_{initial}$ ($\geq$ -3.1 ‰) and higher initial $d$-excess ($\geq$ +10.0 ‰).

In addition, the highest initial $d$-excess ($\geq$ 24.1 ‰) were observed on 2019/11/05 and 2020/09/06 events. A possible explanation of these greater $d$-excess values may be enhanced interaction with the surface of the continent, resulting in evapotranspiration processes. At steady state, transpiration is a non-fractionating process. This means that soil water pumped by plants returns to the atmosphere without any detectable change in its isotopic composition (Cuntz et al., 2007; Flanagan et al., 1991; Dongmann and Nürnberg, 1974). If it is assumed that soil water available to plants has isotopic characteristics equal to the mean values of the two events described, then the water vapor released to the local atmosphere during transpiration will possess identical isotopic signatures. Now, assuming that this water vapor is lifted by convection and then condenses, it is possible to easily calculate the isotopic composition of the first condensate. Assuming an isotopic equilibrium between the gaseous and liquid phases of water in the cloud:

$$\delta_L = \alpha_{eq}(1000 + \delta_V) - 1000 \tag{7}$$

where $\delta_L$ and $\delta_V$ signify delta values of liquid (condensate) and vapor phase, respectively, at isotopic equilibrium, whereas $\alpha_{eq}$ stands for equilibrium fractionation factor. Equilibrium fractionation factors for $^2H$, $^{18}O$ and $d$-excess were calculated using empirical expressions proposed by (Horita and Wesolowski, 1994). The assumed condensation temperature was equal 20 °C and 18 °C (cf. Tdw for 2019/11/05 and 2020/06/09, respectively in Table 1). The calculated isotopic characteristics of the first condensate are equal $\delta^2H$ = +85.1 ‰, $\delta^{18}O$ = +6.6 ‰, $d$-excess = +32.3 ‰ and $\delta^2H$ = +81.0 ‰, $\delta^{18}O$ = +6.5 ‰, $d$-excess = +28.8 ‰, for both respectively events. This example calculation suggests the transpiration process could generate isotopically enriched rainfall and greater $d$-excess.

Thus, these regional processes were imprinted in the initial isotopic composition ($\delta^{18}O_{initial}$ and $d$-excess) of all convective events. This regional $\delta$-signature was preserved during summer (2020/01/30 and 2020/02/01), autumn (2020/06/09) and spring (2019/11/05) events, as indicated by similar $\delta^{18}O_{initial}$, $\delta^{18}O_{med}$, lower $\Delta\delta^{18}O$ values. In addition, the $d$-excess exhibited greater difference between $\delta_{initial}$ and $\delta_{med}$, and higher $\delta\Delta$ values in relation to the $\delta^{18}O$ parameters for all convective rainfall events. The following section provides more detail on the variability of $d$-excess in terms of local atmospheric processes.

**4.2 Local atmospheric controls**

397        The events on summer (2020/02/10 and 2021/02/24), autumn (2020/05/23) and spring (2020/11/18) exhibited substantial

differences in $\delta_{initial}$, $\delta_{med}$, and higher $\Delta\delta$ for $\delta^{18}O$, $\delta^{2}H$ and $d$-excess (Table 1), implying that local processes modified the
regional isotopic imprint.

400        Overall, the Rayleigh distillation governs the depletion of isotopic composition for the events 2020/02/10 ($^{18}O_{initial}$ = -12.3

‰ and $\delta^{18}O_{med}$ = -13.9 ‰) and 2020/11/18 ($^{18}O_{initial}$ = -2.7 ‰ and $\delta^{18}O_{med}$ = -4.1 ‰). This depletion is linked to a reduction of
isotopic exchange and the local increase in cloud-top heights, which leads to a rise in BT values observed at both events,
ranging from -16 to -45 °C (Fig. 5l) and -62 and -65 °C (Fig. 6l), respectively. The intra-event analysis facilitates identification
of variable fractionation processes during the evolution of these rainfall systems. The $\delta^{18}O$ trends of both events show
similarities, but notable differences in $d$-excess trends occur due to varying vertical profiles and RH conditions. On 2020/02/10,
the Zc changed towards the end of the event while RH remained consistently high (97 %). This induced a change in $d$-excess
during a specific time of the event. On the other hand, on 2020/11/18, Zc was varied at the beginning of event with lower RH
of 70 ~ 82 %, leading to a lower $d$-excess during the start of event. The observed strong and significant correlations between
isotopic composition and RH support this variation for both events.

410        The event of 2021/02/24 provides a suitable example of the impact of local factors. The marked differences between the

initial and median values for $d$-excess (13 ‰ and 7.2 ‰, respectively) and the isotopic composition, enriched with initial
($\delta^{18}O_{initial}$ = -7.6 ‰ and $\delta^{2}H_{initial}$ = -47.8 ‰) and median ($\delta^{18}O_{med}$ = -6.8 ‰ and $\delta^{2}H_{med}$ -44.8 ‰) values (Table 1), resulted in a
distinctive enrichment in the isotopic composition. This enrichment is associated with the diverse vertical structure of rainfall
and low humidity conditions (RH, 78 ~ 88%). Alterations in both rainfall patterns and Zc levels under low humidity conditions
promote the preferential escape of lighter isotopologues from liquid water (Dansgaard, 1964). This is corroborated by notable
and negative correlations between isotopic composition, rainfall volume, and Zc. In addition, the preferential escape of lighter
isotopologues also occurred during the 2020/05/23, characterized by lower RH (78 ~ 89 %), resulted in enriched isotopic
composition.

419        The semi-quantitative evaluation illustrated in Table 2 reinforces the intra-event analysis, suggesting a modification of the

mean $d$-excess. The intra-event results indicate that local changes in the isotopic composition of rainfall are controlled by the
specific cloud characteristics and the vertical structure of rainfall, which are connected to local humidity conditions. Therefore,
the reduction in $d$-excess was greater during the events on 2021/02/24, 2020/05/23, and 2020/11/18 due to cloud features and
low humidity conditions, compared to the event on 2020/02/10 that had high local humidity conditions.
**5 Concluding remarks**

425        The study employed high-frequency isotope parameters ($\delta_{initial}$, $\delta_{med}$, and $\Delta\delta$) as well as meteorological data to investigate

the regional and local mechanisms controlling the isotopic characteristics of convective precipitation.
Based on inter-event analysis, it has been revealed that the regional isotopic characteristics are different between summer
and autumn-spring seasons. The $\delta_{initial}$ is determined by moisture transport mechanisms and convection features. The key
factors are progressive rainfall along trajectories and Rayleigh distillation along the moisture transport pathway. The effect of
rainfall along trajectories is pronounced during summer events, associated with the longer moisture transport pathway from
the Amazon forest, which produces depleted heavy isotopes. In contrast, reduced autumn and spring rainfall along trajectories
are associated with the shorter moisture transport pathway from the Atlantic Ocean and southern Brazil, producing enriched
isotope characteristics. This regional δ-signature has been preserved in both summer, autumn, and spring events. Specific
events in autumn and spring with high $d$-excess values were associated with evapotranspiration processes along the moisture
transport pathway, demonstrating how regional convective processes interact with the tropical surface and alter the isotopic
composition.
During the advance of convective rainfall, the regional δ-signature was altered by local effects generated the isotope
variability (large $\Delta\delta$ values), as shown by the intra-event evaluation. The critical local controls are the cloud changes and the
vertical structure of the rainfall. The local controls occur under certain specific conditions of low relative humidity of ambient.
These local mechanisms amplify the discrepancy between the $\delta_{initial}$ and $\delta_{med}$ values, leading to significant $\Delta\delta$ values.
Significant correlations between $\delta^{18}O$, $d$-excess, Zc, and RH, as well as the semi-quantitative evaluation, lend support to the
significance of the vertical structure and relative humidity conditions outlined in this study.
Therefore, the convective rainfall is controlled by an interplay of regional and local factors. The complex and dynamic
conditions of convective rainfall formation across the tropics can be understood using high-frequency analysis. Through
identifying the complexity of the factors that make up the isotopic composition of convective rainfall in the study area, it was
possible to understand why it was so difficult to apply regression models in past studies when using daily data and separation
of rainfall types for the Rio Claro GNIP station.
Although high-frequency rainfall sampling is logistically difficult, we encourage future studies of this type in different
geographical regions across the tropics, to better understand the factors controlling the isotopic composition of convective
rainfall during rainy period. Extensive monitoring of local meteorological parameters and modeling of regional moisture
transport to the rainfall collection site, along with the application of more robust below-cloud models, should accompany such
studies.

**Data availability**

A complete database (isotope characteristics of rainfall as well as selected meteorological parameters characterizing these
events) are available at: https://doi.org/10.17632/kk3gs8zn4s.1 (dos Santos et al., 2023). Monthly GNIP data:
https://www.iaea.org/services/networks/gnip.          GOES-16          imageries          are          available          at:
https://home.chpc.utah.edu/~u0553130/Brian_Blaylock/cgi-bin/goes16_download.cgi. The weather systems are available at:
https://www.marinha.mil.br/chm/dados-do-smm-cartas-sinoticas/cartas-sinoticas                                          and
http://tempo.cptec.inpe.br/boletimtecnico/pt. Reanalysis data are available at:
(https://cds.climate.copernicus.eu/cdsapp#!/search?type=dataset. The Global Modeling and Assimilation Office (GMAO) data
are available at: https://goldsmr4.gesdisc.eosdis.nasa.gov/data/MERRA2/M2T1NXFLX.5.12.4/).
Goddard Earth Sciences Data and Information Services Center (GES DISC) data are available at:
https://disc.gsfc.nasa.gov/datasets/AIRS3STD_7.0/summary.

*Acknowledgment*
FAPESP support for the scholarship provided under the Process 2019/03467-3 and 2021/10538-4 is acknowledged. Durán-
Quesada acknowledges time for analysis and writing provided within UCR C1038 project. The authors acknowledge Troy G.
for English revision.

**Financial support**
This work was funded by grants the São Paulo Research Foundation (FAPESP) under Processes 2018/06666-4, 2019/03467-
3 and 2021/10538-4, and by the International Atomic Energy Agency Grant CRP-F31006.

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

**Table 1.** Summarizing overall convective rainfall events, isotope and meteorological parameters

| Season | | Spring | | Autumn | | Summer | | | |
|---|---|---|---|---|---|---|---|---|---|
| Data | | 2019/11/05 | 2020/11/18 | 2020/05/23 | 2020/06/09 | 2020//01/30 | 2020/02/10 | 2020/02/01 | 2021/02/24 |
| Number of samples | | 21 | 8 | 4 | 12 | 6 | 18 | 5 | 16 |
| Duration | | 82 | 141 | 131 | 96 | 23 | 86 | 18 | 55 |
| δ¹⁸O | Initial | -3.0 | -2.7 | -2.6 | -3.6 | -10.1 | -12.3 | -10.2 | -7.6 |
| | Median | -3.1 | -4.2 | -2.9 | -3.4 | -10 | -13.9 | -10.4 | -6.8 |
| | Δδ | 2.4 | 2.6 | 0.8 | 2.2 | 1.1 | 7.3 | 1.5 | 3.5 |
| δ²H | Initial | 3.4 | -4.6 | -4.6 | -5.2 | -60.1 | -86.6 | -71.0 | -47.8 |
| | Median | 0.8 | -13.7 | -6.9 | -5.6 | -64.4 | -92.0 | -73.5 | -44.8 |
| | Δδ | 16.9 | 9.9 | 8.9 | 11 | 10.5 | 43.1 | 7.4 | 20.9 |
| *d*-excess | Initial | 27.4 | 10.2 | 16.7 | 24.1 | 20.8 | 12.1 | 11.3 | 13.0 |
| | Median | 22.9 | 19.7 | 16.3 | 17.3 | 15.7 | 17.5 | 13.4 | 7.2 |
| | Δδ | 7.1 | 12.8 | 4.0 | 19.2 | 9.5 | 16.6 | 8.4 | 17.2 |
| Automatic Weather Station | Rain rate | 0.4 | 0.2 | 0.1 | 0.3 | 0.4 | 0.5 | 0.6 | 0.5 |
| | RH | 96 | 85 | 87 | 95 | 93 | 97 | 93 | 86 |
| | T | 21 | 20 | 19 | 19 | 23 | 22 | 23 | 21 |
| | Tdw | 20 | 17 | 17 | 18 | 21 | 21 | 21 | 18 |
| | LCL | 146 | 489 | 449 | 168 | 247 | 93 | 253 | 468 |
| Micro Rain Radar | Zc | 46 | 38 | 33 | 42 | 38 | 41 | 39 | 35 |
| | w | 8 | 7.1 | 6.6 | 7.7 | 6.6 | 6.7 | 7.1 | 7.1 |
| GOES-16 | BT | -63 | -63 | -56 | -50 | -53 | -39 | -60 | -51 |

Duration (minutes); Isotopes parameters (‰); Median values of meteorological variables: Rain rate (mm.min⁻¹), Relative Humidity – (RH %), Temperature (T °C), Dew Temperature (Tdw °C), Lifting Condensation Level (LCL meters), Reflectivity (Zc dBZ), Vertical Velocity (m.s⁻¹) and Brightness temperature (BT °C).

**Table 2.** The results of semi-quantitative assessment of the *im*pact of below-cloud processes on the isotope characteristics of convective precipitation

| Rainfall event | $T_{INT}$ [a] (°C) | $RH_{INT}$ [b] (%) | F [c] (-) | ▲d-excess [d] (‰) |
|---|---|---|---|---|
| The 2019/11/05 event<br>$\delta_o$ - isotopic composition of rainfall (‰):<br>$\delta^2H = 0.80$, $\delta^{18}O = -3.11$, *d*-excess = 25.7<br>$\delta_A$ – isotopic composition of equilibrium vapour (‰)[e]:<br>$\delta^2H = -78.3$ $\delta^{18}O = -12.84$, *d*-excess = 24.4 | 19.3 | 97.8 | 0.9982 | 1.7 |
| The 2020/11/18<br>$\delta_o$ - isotopic composition of rainfall (‰):<br>$\delta^2H = -13.7$, $\delta^{18}O = -4.16$, *d*-excess = 19.5<br>$\delta_A$ – isotopic composition of equilibrium vapour (‰):<br>$\delta^2H = -93.2$ $\delta^{18}O = -14.01$, *d*-excess = 18.8 | 19.0 | 92.9 | 0.9795 | 3.1 |
| The 2020/05/23<br>$\delta_o$ - isotopic composition of rainfall (‰):<br>$\delta^2H = -6.9$, $\delta^{18}O = -2.89$, *d*-excess = 16.2<br>$\delta_A$ – isotopic composition of equilibrium vapour (‰):<br>$\delta^2H = -86.6$ $\delta^{18}O = -12.72$, *d*-excess = 15.2 | 18.1 | 93.4 | 0.9806 | 2.8 |
| The 2020/06/09<br>$\delta_o$ - isotopic composition of rainfall (‰):<br>$\delta^2H = -5.5$, $\delta^{18}O = -3.37$, *d*-excess = 21.3<br>$\delta_A$ – isotopic composition of equilibrium vapour (‰):<br>$\delta^2H = -84.8$ $\delta^{18}O = -13.15$, *d*-excess = 20.4 | 19.3 | 97.5 | 0.9978 | 0.2 |
| The 2020/01/30<br>$\delta_o$ - isotopic composition of rainfall (‰):<br>$\delta^2H = -64.4$, $\delta^{18}O = -10.03$, *d*-excess = 15.8<br>$\delta_A$ – isotopic composition of equilibrium vapour (‰)[e]:<br>$\delta^2H = -135.5$ $\delta^{18}O = -19.44$, *d*-excess = 20.0 | 22.4 | 96.4 | 0.9944 | 0.9 |
| The 2020/02/10<br>$\delta_o$ - isotopic composition of rainfall (‰):<br>$\delta^2H = -91.97$, $\delta^{18}O = -13.85$, *d*-excess = 18.8<br>$\delta_A$ – isotopic composition of equilibrium vapour (‰)[e]:<br>$\delta^2H = -161.6$ $\delta^{18}O = -23.28$, *d*-excess = 24.6 | 21.7 | 98.6 | 0.9994 | 0.1 |
| The 2020/02/01<br>$\delta_o$ - isotopic composition of rainfall (‰):<br>$\delta^2H = -73.5$, $\delta^{18}O = -10.44$, *d*-excess = 10.2<br>$\delta_A$ – isotopic composition of equilibrium vapour (‰)[e]:<br>$\delta^2H = -143.8$ $\delta^{18}O = -19.80$, *d*-excess = 14.6 | 22.5 | 96.3 | 0.9947 | 0.9 |
| The 2021/02/24<br>$\delta_o$ - isotopic composition of rainfall (‰):<br>$\delta^2H = -44.8$, $\delta^{18}O = -6.79$, *d*-excess = 9.5<br>$\delta_A$ – isotopic composition of equilibrium vapour (‰):<br>$\delta^2H = -120.3$ $\delta^{18}O = -16.48$, *d*-excess = 11.5 | 19.3 | 93.2 | 0.9800 | 3.0 |

a) mean temperature of below cloud ambient atmosphere (linear interpolation between cloud base and ground level values)

b) mean relative humidity of below cloud ambient atmosphere (linear interpolation between cloud base and ground level values)

c) remaining mass fraction of raindrops after their travel from the cloud base to the surface (see text)

d) reduction of the $d$-excess of raindrops as a result of their travel from the cloud base to the surface (see text)

e) assumed isotopic composition of ambient humid atmosphere below the cloud base derived from the measured isotopic composition of rainfall and ground-level temperature.

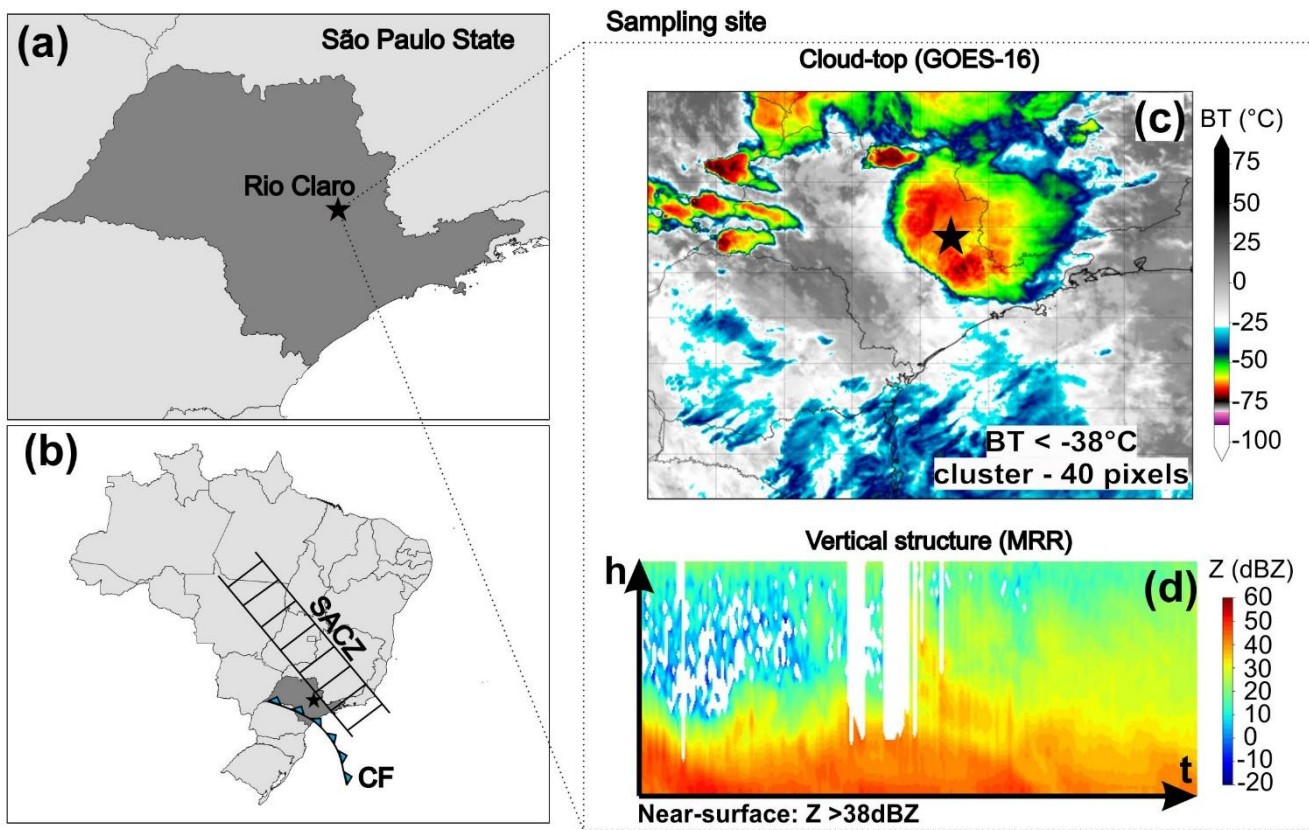

**Figure 1**. Regional and local context of study area. (a) Localization of sampling site in Rio Claro (black star) (b) regional synoptic context across Brazil and main weather systems (CF – cold front and SACZ – Southern Atlantic Convergence Zone). (c) GOES-16 satellite imagery of convective rainfall (d) Micro Rain Radar (MRR) image of convective rainfall.

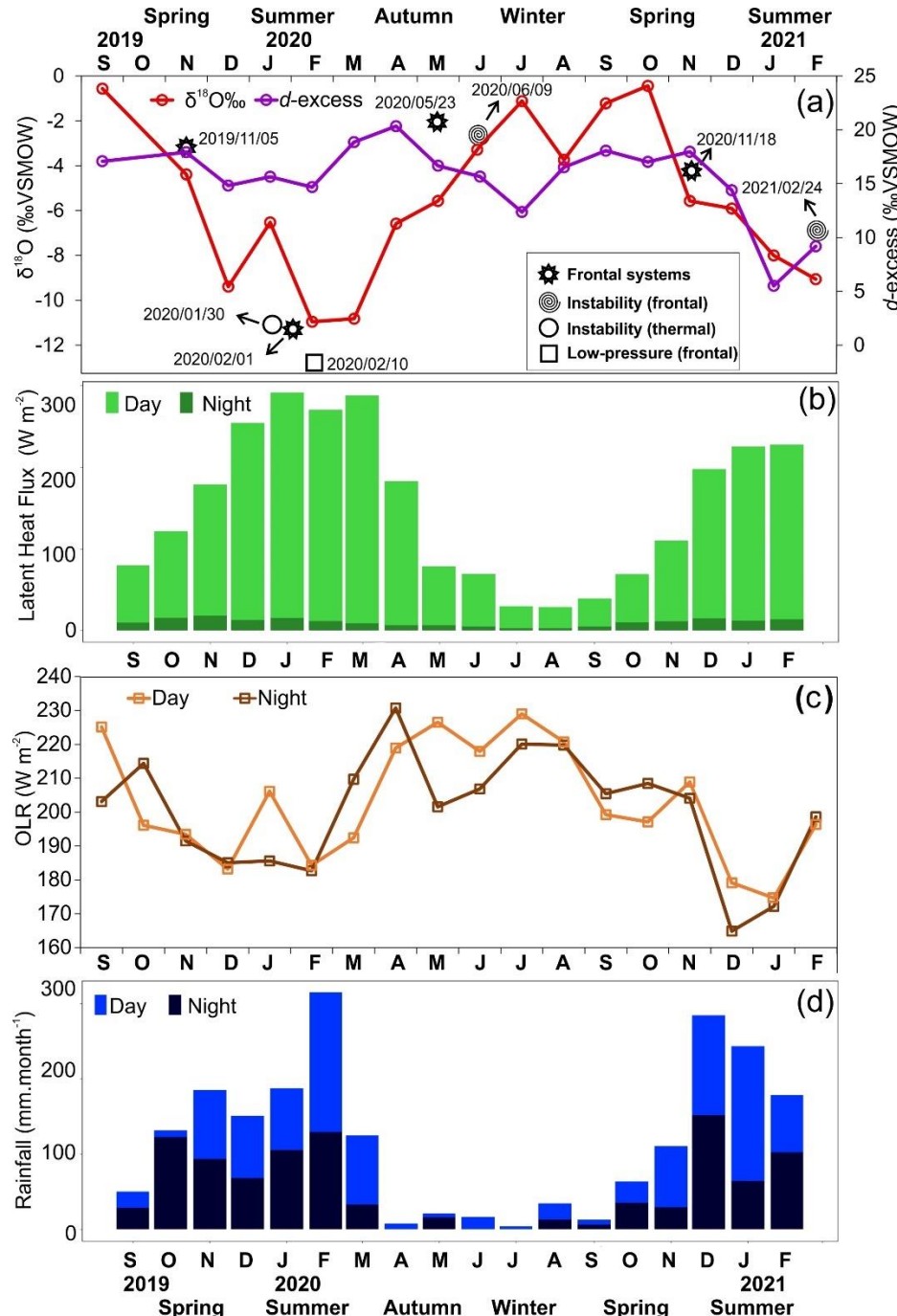

**Figure 2.** Seasonal variation of isotope and convective parameters. (a) Temporal distribution of monthly δ$^{18}$O and *d*-excess values during study period, with aggregated median of δ$^{18}$O values for high-frequency convective rainfall events (b) AQUA/AIRS latent heat flux. (c) MERRA-2 outgoing longwave radiation (monthly averaged daytime and night-time data) (d) monthly rainfall amounts at Rio Claro separated into day and night fraction (no rainfall types distinguished). The black symbol indicates weather systems described in section 3.1. The monthly isotopic composition used in this figure was collected by the first authors of the article and determined by the UNESP laboratory, following the same procedures mentioned in section 2.2.

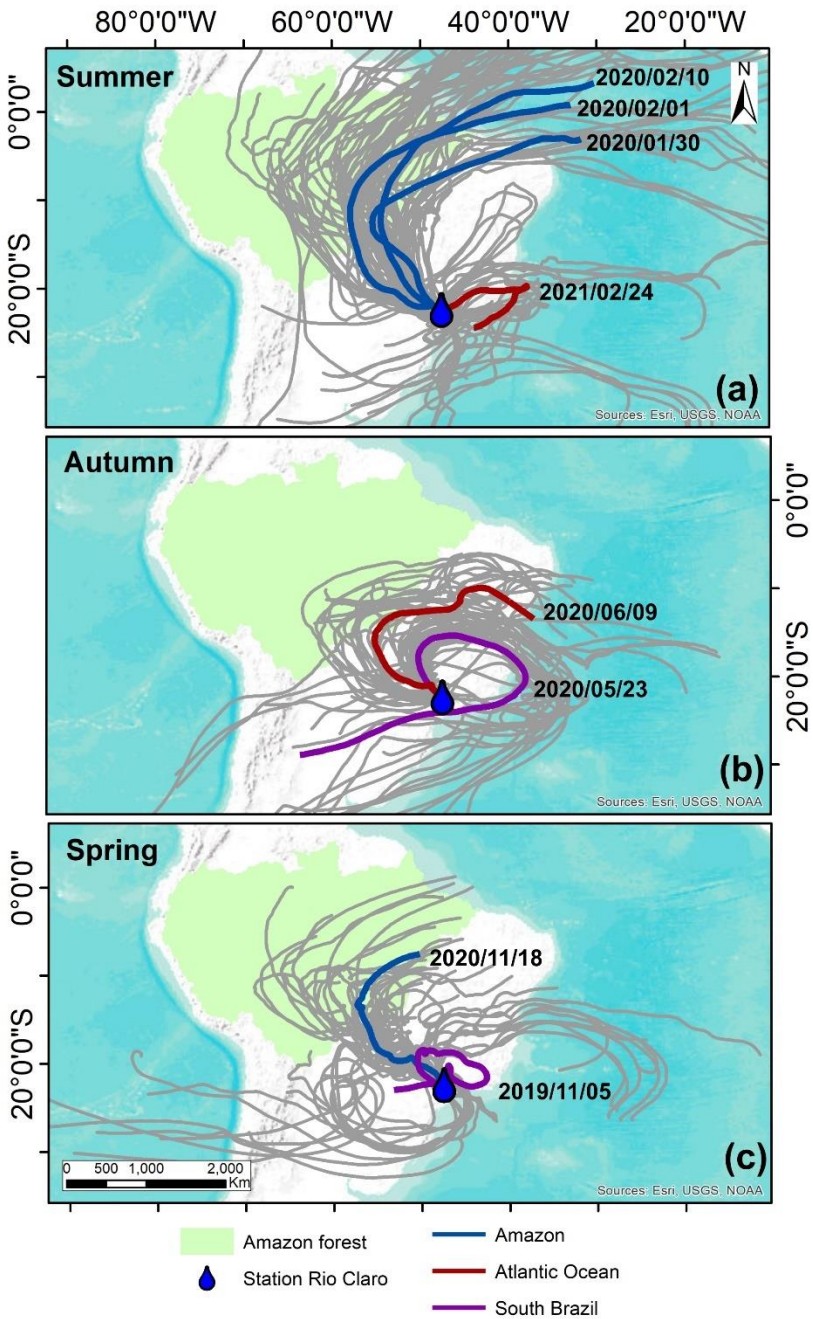

Figure 3. Ten-day backward trajectories arriving at Rio Claro station of eight convective events. (a) Summer, (b) Autumn and (c) Spring. Twenty-seven ensembles are grey lines, and the mean trajectory is the colors lines. The colours of the mean trajectories indicate the origin of air masses. The authors used trivial information, the borders of the countries and the ocean provided by the ESRI base map.

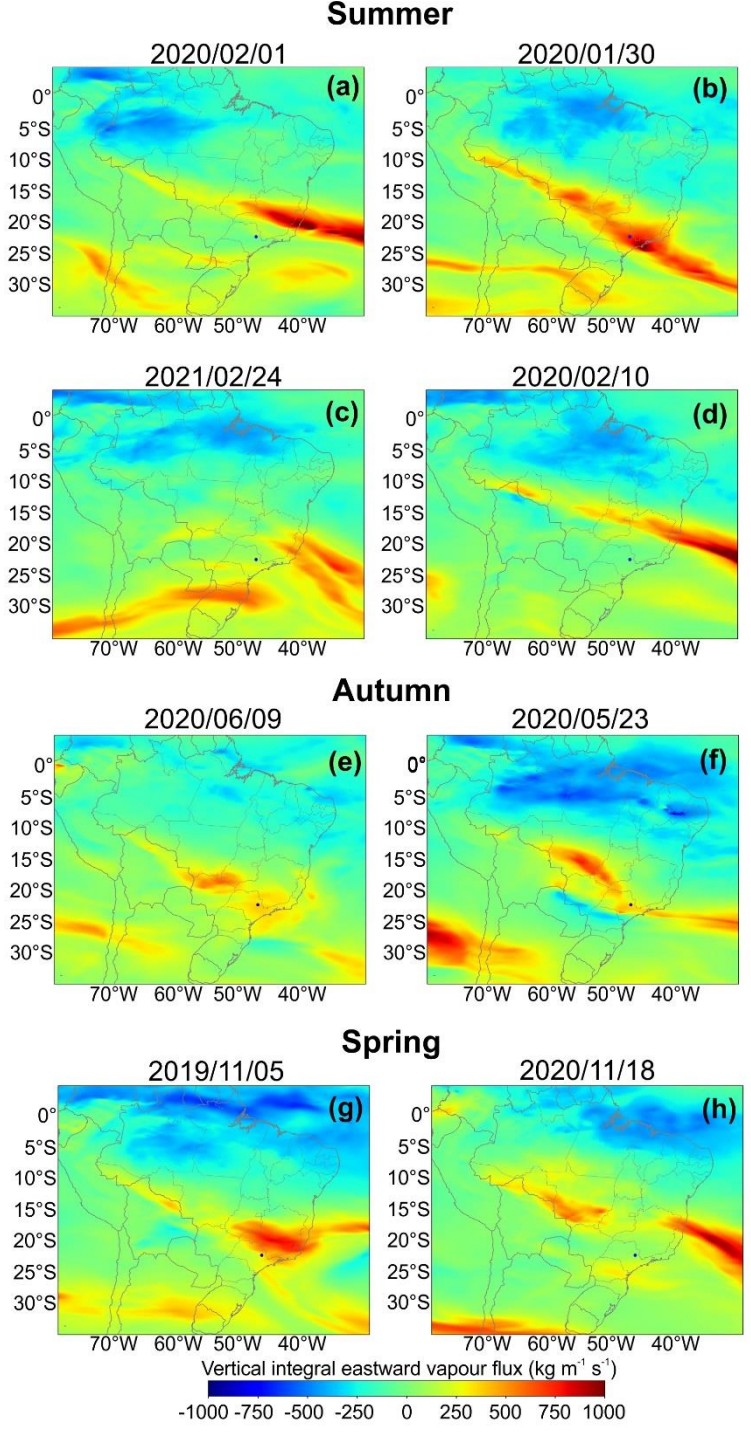

**Figure 4.** ERA-5 vertical integral of eastward water vapor flux. (a, b, c, d) summer convective events (e, f) autumn and (g, h) spring aggregated. The maps corresponded to the days when convective rainfall events occurred. Positive values indicate the direction of moisture vapor flux from left to right, and negative values from right to left.

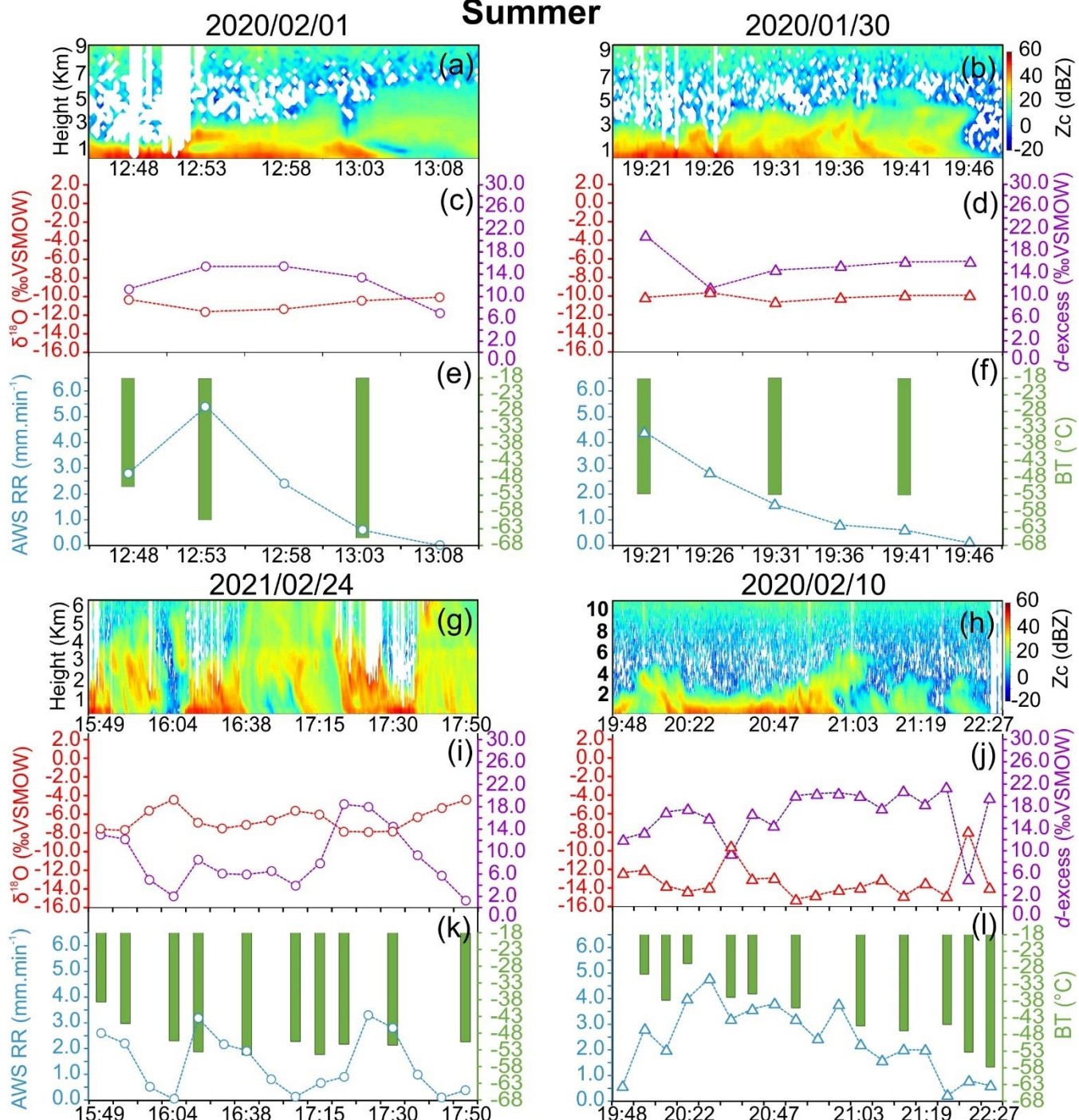

**Figure 5.** Summer intra-events. (a, b, g, h) radar reflectivity of Micro Rain Radar (c, d, i, j) $\delta^{18}O$ (red lines) and $d$-excess (purple lines) (e, f, k, l) brightness temperature (BT – green bars) and rainfall amount (blue lines).

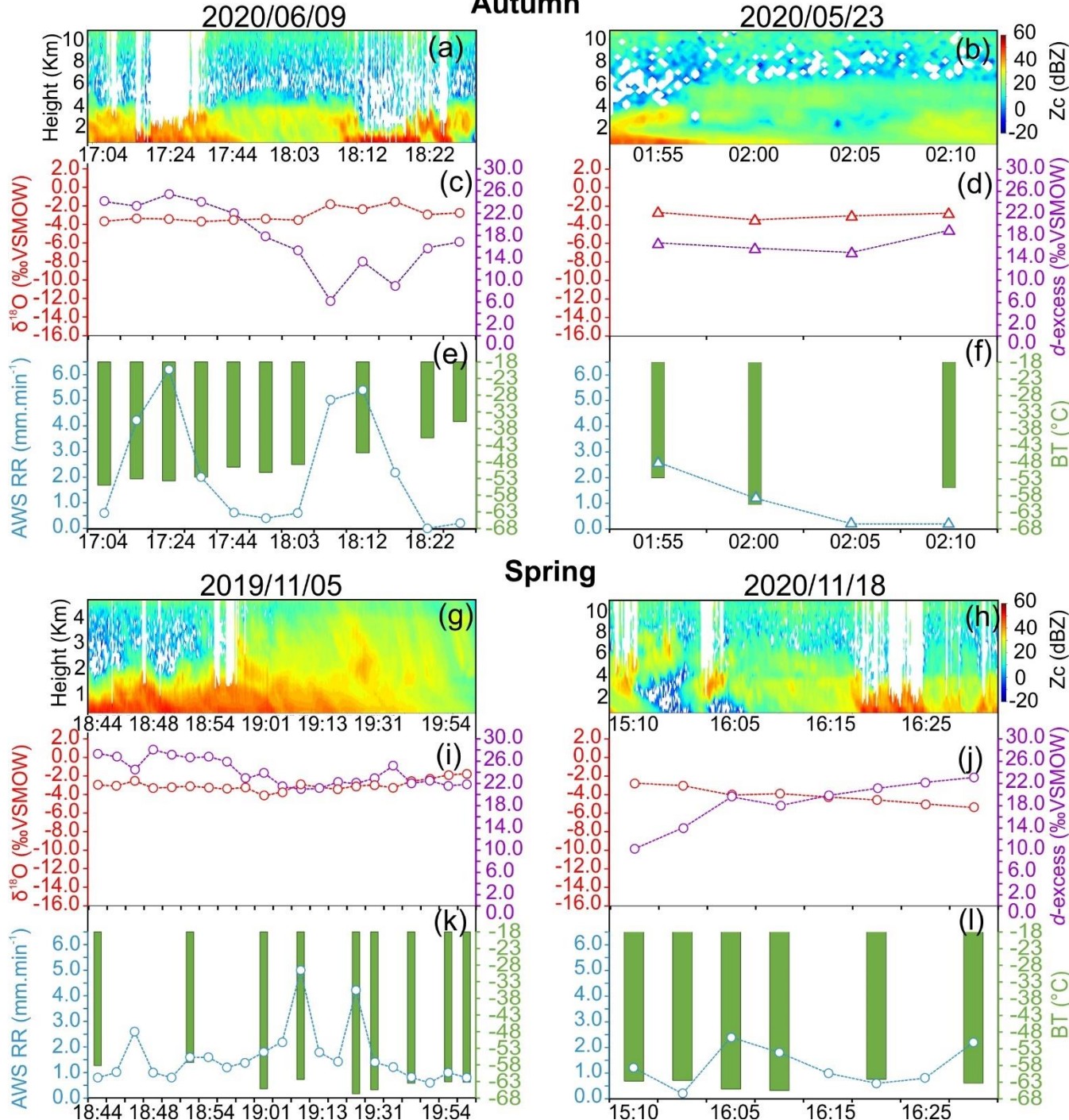

**Figure 6.** Autumn and spring intra-events. Refer to Fig. 5 for legend description.