# Peer review of "Isotopic composition of convective rainfall in the inland tropics of"

_EGUsphere, 2023_

## Referee Comment (RC1)

**Review of dos Santos et al**

May 23, 2023

This article presents measurements in rain water samples collected at a high-frequency in inland tropical Brazil. The article focuses on the day-night contrast between the isotopic composition of the rain, which can mainly be observed in summer. The authors conclude on the importance of sub-cloud processes to explain the day-night contrast. This adds to the growing body of research showing the importance of sub-cloud processes in controlling the water isotopic composition. This paper is significant for the community of people interested in interpreting isotopic measurements for applications both for present-day processes and for paleoclimate reconstructions.

The paper is generally well-written and illustrated. However, although the title focuses on the day-night contrasts, the article presents results at various scales: seasonal, inter-event, intra-event, diel. This mixture of time scales is confusing (major comment). The structure of the manuscript could also be improved. Currently the discussion section is very redundant with the results section and does not bring any new element.

**1 Major comment: Confusion between time scales**

- The article emphasizes the high-frequency sampling of the precipitating systems (5-10 minutes). This should allow the analysis of intra-event variations with precipitating systems. Why aren't these variations analyzed?

- The article focuses on the diel contrast between precipitating systems occurring during the day and during the night. Why is high-frequency sampling useful for this goal? What not just sampling the precipitation twice a day, once at sunrise and once at sunset?

- In Fig 3 and 4, what do the markers represent? Individual samples at the 5-10 minutes scale? Does the variability across markers mix the intra-event variability and the inter-event variability? If so, isn't this confusing to mix the two scales of variability?

- Fig 4 and Table 1 aim at identifying relationships for a given season and for a given period of the day across samples that reflect intra-event and/or inter-event variability. However, most of the discussion focuses on the day-night contrast. So it's not clear how Fig 4 and Table 1 feeds the discussion on the day-night contrast.

- Make different sections in the paper, one devoted to the day-night contrast, and one devoted to the inter-event variability and/or intra-event variability? Alternatively, the paper could be focused on the day-night contrast only.

**2 Minor comments**

- l 43: "low-level convergence (stratiform clouds)": no, stratiform are rather associated with low-level divergence, since there is a mesoscale ascent above the freezing level and mesoscale descent below [Houze, 1989, Houze, 2004]

- l 70: also high-frequency within squall lines, e.g. [Taupin et al., 1997, Risi et al., 2010, Tremoy et al., 2014]

- l 108: clarify errors using the same format as l 107.

- l 108: what is the error on d-excess?

- l 137: "used calculations" -> "used for calculations"

- l 139: "data, for" -> "data were used for"

- l 185-186: confusing: why aren't there any rainfall samples collected at night during spring? Because of Ovid restrictions or because convective events occurred predominantly during the day? What percentage of events are we missing due to covid restrictions?

- l 188-189: unclear logical link: why are there equal amounts of rainfall during the day and nights? Previous studies suggest that this diel distribution mainly depends on the convective organization, types of convective systems and their propagation, e.g. [Tai et al., 2021, Sato et al., 2009]

- l 211: "different meteorological scenarios for" -> "meteorological factors controlling"

- l 226: clarify. It really depends on the meteorological variable. Clearly explain which variable is higher and which is lower.

- l 236-237: Risi et al 2019 and Hu et al 2022 are not appropriate here. The coincidence between LCL and cloud base has been known for many decades

- l 243: "less vigorous" -> "smaller"

- l 244: the logical link between cloud depth and time of interaction of raindrops with the ambient air is not clear. Only the link with the cloud-base altitude makes sense.

- l 249: remove "(not include vapor isotope data)". Because a lot can be said about convective processes with precipitation data, e.g. [Risi et al., 2010]

- l 251-254: unclear. Could be removed.

- l 263: clarify where the wind comes from: e.g. Northwesterly.

- l 277: cite papers documenting the isotopic signature of evapo-transpired moisture, e.g. [Salati et al., 1979, Gat and Matsui, 1991, Risi et al., 2013, Shi et al., 2022]

- l 292: Clarify and/or simplify. If both night-time events and all events come from the Amazon, nothing is different.

- l 294(295: this is not specific to day/night. I recommend to tighten the discussion to demonstrate one or a few key messages. e.g. what explains the day-night contrasts. Anything that is not specific to day or night shouldn't be discussed here.

- l 296: "Therefore": the logical link is not clear. The 2 sentences seem to contradict each other.

- l 305-306: this appears to contradict the rest of the paper and the summary schematic in Fig 5. Basically, there are 2 kinds of processes that can explain the day-night contrast: (1) local processes, (2) regional processes. How do you quantify the relative importance of these 2 kinds of processes? A clear picture should emerge on which kind of process is most important, and it should be cosistent between different parts of the text and the summary schematic.

- l 308: "partial evaporation and isotope exchange of raindrops" -> "partial evaporation of raindrops and rain-vapor interactions"

- l 320-324: clarify this rationale. Previous studies on the diurnal cycle of convection in this region could also be cited to support or refute the rationale.

- Fig 1: for which seasons are the back-trajectories plotted? In the text, different origins depending on seasons are discussed, but this does not appear on this Fig.

- Fig 4p-t: how are the different types of systems defined? It would be useful to add a subsection on this classification in the Methods section.

- Fig 5: why are droplets smaller during the day? This represents stronger rain evaporation, but is it consistent with observations of rain size distribution?

- Sub-section 2.4 on "convective rainfall classification": where is this classification used in the paper? If not used, remove

**References**

[Gat and Matsui, 1991] Gat, J. R. and Matsui, E. (1991). Atmospheric water balance in the Amazon basin: An isotopic evapotranspiration model. *J. Geophys. Res.*, 96:13179–13188.

[Houze, 1989] Houze, R. A. (1989). Observed structure of mesoscale convective systems and implications for large-scale heating. *Quart. J. R. Meteor. soc.*, 115 (487):425–461.

[Houze, 2004] Houze, R. A. (2004). Mesoscale convective systems. *Rev. Geophys.*, 42 (4):DOI: 10.1029/2004RG000150.

[Risi et al., 2010] Risi, C., Bony, S., Vimeux, F., Chong, M., and Descroix, L. (2010). Evolution of the water stable isotopic composition of the rain sampled along Sahelian squall lines. *Quart. J. Roy. Meteor. Soc.*, 136 (S1):227 – 242, DOI: https://doi.org/10.1002/qj.485.

[Risi et al., 2013] Risi, C., Noone, D., Frankenberg, C., and Worden, J. (2013). Role of continental recycling in intraseasonal variations of continental moisture as deduced from model simulations and water vapor isotopic measurements. *Water Resour. Res.*, 49:4136–4156, doi: 10.1002/wrcr.20312.

[Salati et al., 1979] Salati, E., Dall'Olio, A., Matsui, E., and Gat, J. (1979). Recycling of water in the Amazon basin: An isotopic study. *Water Resources Research*, 15:1250–1258.

[Sato et al., 2009] Sato, T., Miura, H., Satoh, M., Takayabu, Y. N., and Wang, Y. (2009). Diurnal cycle of precipitation in the tropics simulated in a global cloud-resolving model. *Journal of Climate*, 22(18):4809–4826.

[Shi et al., 2022] Shi, M., Worden, J. R., Bailey, A., Noone, D., Risi, C., Fu, R., Worden, S., Herman, R., Payne, V., Pagano, T., et al. (2022). Amazonian terrestrial water balance inferred from satellite-observed water vapor isotopes. *Nature communications*, 13(1):1–10.

[Tai et al., 2021] Tai, S.-L., Feng, Z., Ma, P.-L., Schumacher, C., and Fast, J. D. (2021). Representations of precipitation diurnal cycle in the amazon as simulated by observationally constrained cloud-system resolving and global climate models. *Journal of Advances in Modeling Earth Systems*, 13(11):e2021MS002586.

[Taupin et al., 1997] Taupin, J.-D., Gallaire, R., and Arnaud, Y. (1997). Analyses isotopiques et chimiques des précipitations sahélienne de la région de Niamey au Niger: implications climatologiques. *Hydrochemistry*, 244.

[Tremoy et al., 2014] Tremoy, G., Vimeux, F., Soumana, S., Souley, I., Risi, C., Cattani, O., Favreau, G., and Oi, M. (2014). Clustering mesoscale convective systems with laser-based water vapor delta18O monitoring in Niamey (Niger). *J. Geophys. Res.*, 119(9):5079–5103, DOI: 10.1002/2013JD020968.

---

## Author Comment (AC1)

**RESPONSE LETTER**

Prof. Dr. Didier Gastmans

Environmental Studies Center – São Paulo State University

Av. 24A, 1515 – Rio Claro (SP) – Brazil

Rio Claro, August 25th, 2023.

Dear Prof. Thijs Heus

The reviewers' comments were excellent and helped us to improve our work. Thus, we agree with the two reviewers' main questions regarding the sampling interval, structure of the article, rainout history and the below cloud processes. For this reason, we decided to radically change the result presentation and discussion section. In the previous version, we decided to group the rain events together to form a more robust dataset, however this grouping ended the explanation of the dataset as complicated and therefore caused the many doubts mentioned by both reviewers in their comments.

The proposed reorganization of the manuscript in an intra-event and inter-event separated evaluation, explaining how regional processes (mainly those related to moisture transport) and local processes (day and night differences, local evaporation) govern the isotopic variability observed in the 8 convective rainfall events evaluated in this study. The new manuscript structure is detailed below, and the main modifications were:

**1. Title:** We decided to modify the title in order to fit better with the changes made in the manuscript, and not only regarding the day-night differences observed. The new title is "Isotopic composition of convective rainfall in the inland tropics of Brazil".

**2. Structure of the article:** Introduction and methods have preserved the same structure of the previous version, only localized modifications were made according reviewers comments, however we have modified results and discussion sections, that were combined and presented together divided into 5 sub-sections: i) presentation of the general events and meteorological parameters associated; ii) seasonal variation in isotopic composition and meteorological parameters, that helps to illustrate convective activity in the study area; iii) temporal evaluation of intra-events iv) inter-event evaluation related to regional processes v) isotopic fractionation model for evaluating the impact of local evaporation processes. The sub-sections (i) and (ii) were modified to meet the reviewers' comments. Sections (iii), (iv) and (v) are completely new. We believe that this new manuscript structure will bring more clarity and objectivity to the work.

**3. Present and explanation of the database**

The methods section was rewritten to better explain how the sampling was conducted, it is now explicitly mentioned that periods without information correspond to either those in which rainfall events did not occur or periods in which manual sampling was not possible, including the impact of the Covid-19 restrictions to access the sampling premises. We included the information of the hours used for the day-night time separation of the samples and data.

The result and discussion section was rewritten to better explain the intra-event variability. We included the temporal evolution of isotope characteristics ($\delta^{18}O$, *d*-excess) and selected meteorological parameters (brightness temperature, MRR reflectivity and rainfall amount) of 8 convective rainfall events sampled.

**4. HYSPLIT analysis**

As suggested by one of the reviewers, we have modified the methodology for evaluating the Hysplit trajectories. In this new version, we have estimated the average trajectory based on 27 trajectories calculated using the ensembles module. This modification was detailed in the methods section. Despite the moisture origin and transport has not changed much from the previous version, a new map was generated and presented in a figure.

In result and discussion section, the Hysplit trajectories were related to the inter-event variability of the isotope and meteorological parameters. In addition, the estimated vertical integral eastward vapor fluxes using data from ERA-5 were combined with Hysplit to improve the analysis of moisture origin, recycling and transport.

**5. Below-cloud evaporation processes:** This section is prepared to replace the conceptual model presented in the previous version and improve the reviewer's suggestions about local evaporation processes. In the result and discussion section, an assessment of the semi-quantitative impact of below-cloud processes was computed on the isotope data. The assessment was made on two rainfall events sampled in summer (2020/02/10-night and 2021/02/24-day) to characterize the differences between day and night situations. The modeling assessment was based on previous relevant isotope works (e. g. Craig and Gordon, 1965; Steward, 1975; Gonfiantini, 1986; Horita and Wesolowski (1994); Horita et al., 2008) and revealed the degree of partial evaporation of raindrops below the cloud base.

**6. Minor comments**

As a large part of the article has been modified, it was not necessary to respond to some of the reviewers' minor comments. Nevertheless, all the changes to the text and new references suggested for addition to the article were accepted.

**7. Tables and Figures**

Tables 1 and 2 are new. Table 1 summarizes the sampling of convective rainfall events, and shows for each event the corresponding season, period of daytime, date, number of samples, duration and median values of isotope and meteorological parameters. Table 2 shows the results of the semi-quantitative assessment of the impact of below-cloud processes on the isotope characteristics of convective precipitation.

Figure 1 is the same as the previous version, except the HYSPLIT map was removed. Figure 2 is the same as the previous version. Figures 4 to 6 are new. Figure 4 shows the intra-event variability of all convective rainfall. The temporal evolution of 18O, d-excess, rain rates, brightness temperature (GOES-16 image) and Micro Rain Radar reflectivity plotted in a vertical profile. Figure 5 is a map of ten-day backward trajectories separated into seasons, showing the mean trajectory for each rainfall event. Figure 6 is the ERA-5 vertical integral of eastward water vapor flux for the days when rainfall events occurred.

---

## Author Response (AR1)

**RESPONSE LETTER**

Prof. Dr. Didier Gastmans

Environmental Studies Center – São Paulo State University

Av. 24A, 1515 – Rio Claro (SP) – Brazil

Rio Claro, September 01st, 2023.

Dear. Dr. Thijs Heus,

Detailed point-by-point responses to all referee comments and specify all changes in the revised manuscript are presented below. These referee comments were essential to improve the original manuscript and have contributed for the radical modifications made in the manuscript revised version.

**Specific response to reviewer 1**

**Major comments**

**Comment:** Why aren't these intra-events variations analyzed? Why is high-frequency sampling useful for this goal? What not just sampling the precipitation twice a day, once at sunrise and once at sunset?

**Response:** According to your suggestion, we modified the explanation of the sampling, see lines 90-98 (in subsection 2.2) and included an intra-event (subsection 3.2) and inter-event comparison (subsection 3.3). In addition, we didn't imagine building a work based on day-night differences, we wanted to analyze differences in rainfall types using high-frequency sampling, so these results surprised us as well. We did not sample one event during the day and another during the night on the same day, it was performed in different days (see in detail in Table 1). In this new version, we have used only one convective event collected during the day and one convective event collected at night to present the day-night differences in isotopic composition of rainfall (subsection 3.4).

**Comment:** Fig 4 and Table 1 aim at identifying relationships for a given season and for a given period of the day across samples that reflect intra-event and/or inter-event variability. However, most of the discussion focuses on the day-night contrast. So, it's not clear how Fig 4 and Table 1 feeds the discussion on the day-night contrast.

**Response:** This figure and table were removed, and the discussion was reformulated as presented in the previous response.

**Comment:** Make different sections in the paper, one devoted to the day-night contrast, and one devoted to the interevent variability and/or intra-event variability? Alternatively, the paper could be focused on the day-night contrast only.

**Response:** Thanks for your suggestion. It was accepted. In the new version we focus on two parts: a) local aspects, for characterizing the seasonal distributions, monthly isotopes, outgoing longwave radiation, and latent heat flux were presented in subsection 3.1; an intra-event analysis based on the temporal evolution of isotopic composition and meteorological data (at the surface, the vertical profile of radar and GOES-16), see subsection 3.2. b) regional aspects, using an inter-event comparison with moisture transport and origin by the Hysplit model and reanalysis data, see subsection 3.3.

**Minor comments**

**Comment:** "low-level convergence (stratiform clouds)": no, stratiform are rather associated with low-level divergence, since there is a mesoscale ascent above the freezing level and mesoscale descent below [Houze, 1989, Houze, 2004]

**Response:** The sentence was corrected, and references included. See line 42.

**Comment:** also high-frequency within squall lines, e.g. [Taupin et al., 1997, Risi et al., 2010, Tremoy et al., 2014].

**Response:** These references were included. See line 67.

**Comment:** clarify errors using the same format as l 107. what is the error on $d$-excess?

**Response:** The sentence was added, see lines 114-115.

**Comment:** "used calculations" -> "used for calculations"

**Response:** The sentence was corrected. See lines 157 and 164.

**Comment:** "data, for" -> "data were used for"

**Response:** The sentence was corrected. See line 159.

**Comment:** confusing: why aren't there any rainfall samples collected at night during spring? Because of Covid restrictions or because convective events occurred predominantly during the day? What percentage of events are we missing due to covid restrictions?

**Response:** There aren't rainfall samples collected at night during spring because of the Covid-19 restrictions. As the sampling was performed manually, it was impossible to sample all rainfall events, which difficult the computed a percentage. In addition, the person responsible for the sampling was waiting for the rain considering the university restrictions due to the pandemic. The covid-19 has restricted access to the university, especially at night. The sentence was rewritten, and the explanation of the sampling was detailed in lines: 90-98.

**Comment: 188-189:** unclear logical link: why are there equal amounts of rainfall during the day and nights? Previous studies suggest that this diel distribution mainly depends on the convective organization, types of convective systems and their propagation, e.g. [Tai et al., 2021, Sato et al., 2009].

**Response:** We computed the rainfall amount for day and night only during the monitoring period (2019-2021) to characterize the role of convection and its influence on rainfall. The diurnal cycle in Amazon (Tai et al., 2021) is quite different than the observed in our study area (e. g. higher temperature) because is modulated by different meteorological systems (cold fronts have few influences in Amazon, for instance). The role of the cold front in the study area could contribute to an equal amount of rainfall during day and night, despite the cold front also influencing to organization of the convection across south-eastern portion of Brazil. A major temporal evaluation of convection processes is necessary to compare these other studies, which is not the objective of this article.

**Comment:** Risi et al 2019 and Hu et al 2022 are not appropriate here. The coincidence between LCL and cloud base has been known for many decades.

**Response:** This paragraph and the references used were removed.

**Comment:** the logical link between cloud depth and time of interaction of raindrops with the ambient air is not clear. Only the link with the cloud-base altitude makes sense.

**Response:** The processes below-cloud were entirely rewritten. See subsection 3.4.

**Comment:** remove "(not include vapor isotope data)". Because a lot can be said about convective processes with precipitation data, e.g. [Risi et al., 2010]

**Response:** It was removed.

**Comment:** clarify where the wind comes from: e.g. Northwesterly "Moist air masses of Atlantic origin are transported westward over the Amazon Forest, undergoing intensive recycling and rainout.

**Response:** This sentence was entirely modified. Description about Amazon transport are presented in lines 307-313.

**Comment:** cite papers documenting the isotopic signature of evapo-transpired moisture, e.g. [Salati et al., 1979, Gat and Matsui, 1991, Risi et al., 2013, Shi et al., 2022].

**Response:** The references were added.

**Comment:** Clarify and/or simplify. If both night-time events and all events come from the Amazon, nothing is different.

**Response:** Corrected. It was clarified in lines 256-258.

**Comment:** this is not specific to day/night. I recommend to tighten the discussion to demonstrate one or a few key messages. e.g. what explains the day-night contrasts. Anything that is not specific to day or night shouldn't be discussed here.

**Response:** The structure of the article was modified. The day-night contrast was discussed in only one subsection 3.4, using two convective rainfalls as case studies.

**Comment:** "Therefore": the logical link is not clear. The 2 sentences seem to contradict each other.

**Response:** These sentences were removed.

**Comment:** this appears to contradict the rest of the paper and the summary schematic in Fig 5. Basically, there are 2 kinds of processes that can explain the day-night contrast: (1) local processes, (2) regional processes. How do you quantify the relative importance of these 2 kinds of processes? A clear picture should emerge on which kind of process is most important, and it should be cosistent between different parts of the text and the summary schematic.

**Response:** The day-night schematic in Fig 5 was removed. However, we present new results and discussion about day-night differences in subsection 3.4. For this purpose, we have evaluated two summer events as case studies, for day and night situations. It was computed the influences of below-cloud evaporation processes (major contribution during day-event), consequently, regional processes had a major contribution during the night event. The

mechanism to control isotopic composition in regional processes as detailed in subsection 3.3. This evaluation was not computed for all summer convective events because the other events were different in relation to the duration and meteorological data (Table 1). In addition, differences in day-night during autumn and spring were not observed during the monitoring period (see subsection 3.1) due to the lower convective activity in relation to the summer.

**Comment:** "partial evaporation and isotope exchange of raindrops" -> "partial evaporation of raindrops and rain-vapor interactions"
**Response:** The sentence was removed.

**Comment:** clarify this rationale. Previous studies on the diurnal cycle of convection in this region could also be cited to support or refute the rationale.
**Response:** Previous studies were cited.

**Comment:** Fig 1: for which seasons are the back-trajectories plotted? In the text, different origins depending on seasons are discussed, but this does not appear on this Fig.
**Response:** A new figure of back-trajectories was plotted subdivided into seasons. See Fig. 5.

**Comment:** how are the different types of systems defined? It would be useful to add a subsection on this classification in the Methods section.
**Response:** This information was included in the Methods, see lines 135-140.

**Comment:** why are droplets smaller during the day? This represents stronger rain evaporation, but is it consistent with observations of rain size distribution?
**Response:** In the previous version, data on drop size was not available. Such as for the new version, in this way, which difficult to compute the difference between day and night events. Alternatively, the evaporation processes in raindrops were assessed using the widely conceptual framework for isotope effects, see subsection 3.4.

**Comment:** Sub-section 2.4 on "convective rainfall classification": where is this classification used in the paper? If not used, remove
**Response:** It was modified in subsection 2.6, where was described how the convective rainfall was characterized using micro rain radar and GOES-16 imagery.

**Response to reviewer 2**

**Major comments**

**Comment:** To verify a hypothesis (post-condensation effect and precipitation history), it must be tested quantitatively, not qualitatively.

**Response:** The revised version included the quantitative information to verify the hypothesis of the article. For the post-condensation effect (see subsection 3.4) a semi-quantitative assessment of the impact below-cloud evaporation was carried out using isotope and meteorological values, the main results of this subsection are synthesized in Table 2.

Hysplit ensemble analysis (Fig. 5), ERA-5 eastward vapor flux (Fig. 6) and evapotranspiration values were used to explain the precipitation history between seasons (summer, autumn and spring), see subsection 3.3. For vapor flux and evapotranspiration were included the quantitative values to differentiate the moisture transport. This meteorological dataset was combined with isotopic composition (median values in Table 1) and it was carried out an inter-event analysis.

**Comment:** [...] explanation about sampling intervals and how many precipitation events are used to calculate the daily and nightly averages.

**Response:** This comment has contributed for improving the new version of the manuscript. In the previous version, we decided to group the rain events together to form a more robust dataset. However, for the new version, we modified entire the results and discussion and presented an intra-event analysis to better explore the high-frequency samples. Table 1 and Fig. 4 illustrate the number events and this temporal evolution, respectively. Now, for the day-night contrast (described in subsection 3.4), it was compared one-day (24/02/2021, n = 16) and one-night (2020/02/10, n = 18) summer events with a similar number of samples, respectively.

**Minor comments**

**Comment:** I can't find a description of how many precipitation events are used to calculate the daily and nightly averages.

**Response:** This description was included in the new version in section methods and Table 1, see lines 90-98. A total of 8 convective events have been collected, 4 in summer (2 in the day and 2 nights), 2 in autumn (1 day and 1 night), and 2 in spring (only day). One summer-event during the day and one summer-event at night were used to characterize diurnal differences in subsection 3.4.

**Comment:** I can't understand how many trajectories are used to identify the moisture origin for each precipitation event. What time did you launch the trajectory analysis? How many trajectories do you compute? A single trajectory analysis does not provide reliable moisture source for each precipitation event. And it is necessary to consider the replenished moisture from the surface during the transport to discuss the moisture sources.

**Response:** According to your suggestion, the Hysplit model was modified in relation to the previous version of the manuscript. In the new version, 27 trajectories were used to identify the moisture origin for each rainfall event, see lines 143-153 and Fig. 5. The time was the local start time of each rainfall event.

**Comment:** Introduction of a conceptual model is not a "Results". Results just show your observed results. Your speculation and conceptual model should be noted in Discussion section. The manuscript should be reorganization before resubmission.

**Response:** The conceptual model was removed. We present new results and discussion about the local processes in subsection 3.4. In this subsection, a semi-quantitative assessment of the impact of below-cloud effects was carried out based on the generally accepted conceptual framework for isotope effects, which resulted in lower speculation about evaporation processes than the previous version.

**Comment:** The same reference was cited with different names. For example, Aemisegger et al. 2015a and Aemisegger et al. 2015b refer to the same paper. Similar mistakes is found for Kurita et al. 2013a and Kurita et al. 2013b. Check the reference carefully.

**Response:** The references were revised.

---

## Referee Report (RR1)

**Review of dos Santos et al**

September 22, 2023

The article has been improved relative to the previous version. However, there are still several issues, especially related to the writing that could be shorter and more focused.

In addition to the comments below, I would advice the authors to have their article proof-read by an English-speaking person.

**1 Minor comments**

- l 23-24: "Combining..." is a very vague sentence. Prefer sentences that convey information on what these processes are and how they play.

- l 24: remove "While".

- l 31-32: "Our results..." this sentence is a generalization that sounds a bit presumptuous. All the drivers that are considered in this study have already been explored in previous studies in other regions. I would remove.

- l 52: Kurita 2013 used in-situ observations, not satellite observations.

- l 57-64: is this paragraph really necessary? The diel cycle is not a focus of the study anymore. This looks like a remnant of the previous version.

- section 2: according to the guidelines, all the links can be moved to a section called "Data availability" just before the Acknowledgments.

- l 174: "Discussion" -> "discussion"

- l 188: Fig 3 is not really used in the paper. Remove the figure and this sentence?

- l 199: grammar issue: The -> This?

- l 204: grammar issue: try "despite no relationship being observed"?

- l 204: "a change in" -> "of"?

- l 205-208: unclear sentence: try "there was similar duration, temporal $\delta^{18}O$ evolution and rain rates"

- l 210: grammar issue. Cut sentence: "This illustrates..."

- l 212-216: sentence is too long.

- General comment on the description of the figures: this is very difficult and long to read. The description should be made much shorter. You don't need to give all the values, because we can see it on the Figures. I advice to really focus on what is the main point that you are trying to make. Identify the main results and interpret it.

- l 226-227: does this correspond to the stratiform zone of the convective system?

- l 229-230: simplify sentence, e.g. "RR and BT did not..."

- l 231: grammar issue: subject missing.

- l 239: Fig 6 is called before Fig 5?

- l 239: grammar issue: verb missing

- l 243-247: sentence is too long. Shorten, cut.

- l 248: grammar issue: try "different from those observed"

- l 250: "intra events" -> "events"

- l 251: "control of" -> "on"

- l 252: along the seasons -> depending on the season

- l 252-254: Vague and useless sentence, remove.

- Section 3.3: This is very long and messy. Paragraph l 256-261 and 288-291 could be merged. At the end of l 270 or l 287, we wonder what is the consequence on isotopes. l 298-302 need to be demonstrated. To make this section easier to read, I advice to make one paragraph per season. And for each season, use back-trajectories and meteorological conditions to interpret isotopic variations. The interpretation of isotopic variations should directly follow the description.

- l 325-330: clarify that this is an extreme case. If we assumed that all the vapor comes from precipitation at each recycling step, then after n recycling step, we would have $R_v = \alpha_{eq}^n \cdot R_{v0}$, which is unrealistic.

- l 355: this assumes that the sub-cloud layer is well-mixed. You can probably cite a paper to justify this assumption.

- l 375: why not simply using the kinetic fractionation from Stewart 1975: $\alpha_K = (D/D_{iso})^n$

- l 384: isn't 1mm a bit large for a raindrop? Are there any previous studies showing disdrometer observations in this region, that could help justify this value?

- l 393: "cloud level" -> "cloud base"

- General section 3.4:

  - what value was used for $\delta_A$? How was it chosen?
  - How were the 2 events selected? Because they show extreme conditions? Or to contrast day and night? Clarify.

- l 408-410: I think this was already known, already before this study. I don't think it's fair to say that isotopes variations allowed to demonstrate this. Rather, isotope variations are consistent with atmospheric dynamics in this region that have already been known for a long time. Previous studies describing the atmospheric dynamics and moisture origin for this region can be cited. e.g. for moisture origin: [van der Ent et al., 2010, Gimeno et al., 2012, Zemp et al., 2014].

- l 413: "It's hard to overestimate": strange and vague sentence. Rather, cite previous studies on the impact of deforestation to make a more precise statement.

- l 416: "reduction in precipitation amount": there is an extensive literature on the impact of deforestation on South American rainfall, and results are not always obvious and consistent. These previous studies should be cited to make a more precise statement.

- l 422: "high-resolution": it's not clear to me what was the added value of the high-resolution sampling. What conclusion couldn't have been drawn if you had done only event-scale sampling? This needs to b clarified.

- Fig 4: This fig is too small, it's hard to read. Maybe cut it into 2 pieces?

- Table 2: why discussing only the impact on d-excess, an not also on deltas?

**References**

[Gimeno et al., 2012] Gimeno, L., Stohl, A., Trigo, R. M., Dominguez, F., Yoshimura, K., Yu, L., Drumond, A. R. D. M., Duran-Quesada, A. M., and Nieto, R. (2012). Oceanic and terrestrial sources of continental precipitation. *Rev. Geophys.*, 50(4):doi:10.1029/2012RG000389.

[van der Ent et al., 2010] van der Ent, R. J., Savenje, H. H. G., Schaefli, B., and Steele-Dunne, S. C. (2010). Origin and fate of atmopheric moisture over continents. *Water Resour. Res.*, 46:W09525.

[Zemp et al., 2014] Zemp, D., Schleussner, C.-F., Barbosa, H., Van der Ent, R., Donges, J. F., Heinke, J., Sampaio, G., and Rammig, A. (2014). On the importance of cascading moisture recycling in south america. *Atmospheric Chemistry and Physics*, 14(23):13337–13359.

---

## Referee Report (RR2)

I have completed my review of the manuscript by dos Santos et al. In their study, the authors collected rain samples at min intervals during several rain events between 2019-2020 and analyzed these samples for stable isotopes. Using these isotopic data in conjunction with meteorological parameters, OLR and reanalysis data, the authors explored the local and regional factors controlling precipitation isotope in their study area. Although the topic is undoubtedly interesting and has significant implications for understanding rain isotopes in the region, there is room for improvement in both language and scientific aspects before this manuscript can be accepted for publication.

Firstly, I found that this article is not easy to read, partially because of language-related issues. For instance, there are grammatical problems in some sentences. An example from the abstract is: "While convective activity, associated with outgoing longwave radiation (OLR) and moisture transport, evaluated from Hysplit modelling and ERA-5 eastward vapor flux, modulate the seasonal rainwater isotopic composition." This sentence seems to be a compound sentence, and the question arises as to what the subjects of these sentences are in this compound sentence. Additionally, there are several overly long sentences that could split into two for improved readability. For example, the sentence between lines 41-44 consists of more than 60 words.

Secondly, there are some organization issues that need attention. The detailed explanation about the models (Lines 355-390) should be included in Data and Method section but not in the Discussion. Also, in this article, the isotope values of vapor in equilibrium with rainwater were mentioned a few times in the main text and tables. The authors should indicate how they calculated these equilibrium isotope values for vapor in Data and Method section.

Furthermore, I recommend that the authors leverage the high-resolution isotope data along with other valuable meteorological parameters available (e.g., MRR data) to explore the evolution of rain isotopes during convective events and its controls rather than devoting a significant portion (Section 3.3) of the text to discussing the correlation of inter-event variability of isotopes with meteorological parameters. It seems to me that the similar conclusions could be reached using daily rainwater isotope data without the need of high-resolution event data at min interval. In addition, waterline defined by isotopes of rainwater from each event can also provide insights into below-cloud evaporation.

Regarding the sampling strategy, I understand that the lockdown indeed affect the rain sapling on university campus due to limited access. The authors used stable isotope of two or three rain events collected during the daytime to represent the isotopes of day events in their area and two or three events collected during the night-time to represent night-time events. Moreover, these events were collected in quite different and in different years. The differences observed between stable isotopes of daytime and night-time rainwater samples collected in different dates, particularly different years, might be due to the inter-annual variability. Therefore, the sampling strategy in this study cannot answer the question raised in the introduction section about the diurnal variation in precipitation in their study area, as diurnal variation refers to variation over 24 hours in a day. The statement in the abstract, "A semiquantitative evaporation model evaluated local influences in summer convective events revealing distinct isotope characteristics between day (high $\delta^{18}O$, low d-excess and substantial evaporation) and night (low $\delta^{18}O$, high d-excess and negligible evaporation)", can be easily challenged by their own data presented in Figure 4. Clearly, events from 2020/02/01 (day event) and 2020/01/30 (night event) were collected in the same year with collection dates only one day apart. These two events exhibits pretty similar isotopic characteristics, evolution pattern, and meteorological parameters.

Finally, a few papers cited in the main text are not in the list of the references at the end of the article. There are also some formatting issues with the papers in the list.

Please refer to the attached annotated PDF file for more detailed comments and suggestions.

[revised manuscript text omitted]

$\delta_o$ - isotopic composition of rainfall (‰):
$\delta^2H$ = -44.8, $\delta^{18}O$ = -6.79, $d$-excess = 9.5
$\delta_A$ – isotopic composition of equilibrium vapour (‰):
$\delta^2H$ = -120.3 $\delta^{18}O$ = -16.48, $d$-excess = 11.5 | 19.3 | 93.2 | 0.9800 | 3.0 |

a) mean temperature of below cloud ambient atmosphere (linear interpolation between cloud base and ground level values)
b) mean relative humidity of below cloud ambient atmosphere (linear interpolation between cloud base and ground level
values)
c) remaining mass fraction of raindrops after their travel from the cloud base to the surface (see text)
d) reduction of the d-excess of raindrops as a result of their travel from the cloud base to the surface (see text)
e) assumed isotopic composition of ambient humid atmosphere below the cloud base derived from the measured isotopic
    composition of rainfall and ground-level temperature.

[Figure]

**Figure 1**. (a) Localization of sampling site in Rio Claro (black star) (b) in regional synoptic context across Brazil and main weather systems (CF – cold front and SACZ – Southern Atlantic Convergence Zone). Over collection point (c) GOES-16 satellite imagery showed convective system with lower brightness temperature (BT, clou-top) and (d) Micro Rain Radar (MRR) illustrates the vertical structure of convective rainfall, height (h) and time (t), characterized by radar reflectivity (Z) with strong values near-surface.

[Figure]

**Figure 2.** (a) Seasonal variation of δ¹⁸O and *d*-excess values in monthly rainfall and aggregated monthly δ¹⁸O values high-frequency convective rainfall sampling discussed in this study (b) AQUA/AIRS latent heat flux. (c) MERRA-2 outgoing longwave radiation (monthly averaged daytime and night-time data) (d) monthly rainfall amounts at Rio Claro separated into day and night fraction (no rainfall types distinguished). The star symbol indicates the collected high-frequency events, ranked according to average values of δ¹⁸O.

[Figure]

**Figure 3.** (a) Monthly and (b) high-frequency $\delta^2H$ and $\delta^{18}O$ rainfall data plotted in the $\delta^2H/\delta^{18}O$ space. LMWL – local meteoric water line based on monthly values, CMWL – convective meteoric water line and GMWL – global meteoric water line.

[Figure]

**Figure 4.** Intra-event variability of eight convective rainfall sampling. For summer season, 2020/02/01(a, c, e), 2020/01/30 (b, d, f), 24/02/21 (g, i, k) and 2020/02/10 (h, j, l), autumn, 09/06/2020 (m, o, q) and 23/05/2020 (n, p, r), autumn 05/11/2019 (s, u, w) and 18/11/2020 (t, v, x). δ¹⁸O is red color, *d*-excess is orange, rain rate (RR) in dark blue, brightness temperature (BT) in green. The Zc is the corrected reflectivity of Micro Rain Radar plotted as vertical profile.

[Figure]

**Figure 5.** Ten-day backward trajectories arriving at Rio Claro station of eight convective events on (a) Summer, (b) Autumn and (c) Spring. Twenty-seven ensembles are grey lines, and the mean trajectory is the colors lines. The colours of the mean trajectories indicate the origin of air masses: blue influenced by Amazon Forest, Tuscan red from Atlantic Ocean and Purple from South Brazil portion. Symbols are daytime of convective events, day (circle) and night (triangle). The authors used trivial information, the borders of the countries and the ocean provided by the ESRI base map.

[Figure]

**Figure 6.** ERA-5 vertical integral of eastward water vapor flux for the days when convective rainfall events occurred, during (a, b, c, d) summer, (e, f) autumn and (g, h) spring aggregated with weather systems text. Positive values indicate the direction of moisture vapor flux from left to right, and negative values from right to left. Arrows illustrate the direction of vapour flux. The weather systems are indicated for each rainfall event.

---

## Referee Report (RR3)

**Review of dos Santos et al**

**December 16, 2023**

The paper has been revised, although the responses to comments are minimal. However, there is still significant work on this paper before it can be acceptable. Through the revision process, the authors improve some aspects and deteriorate others.

There are still big issues with the language. I advice a thorough proof-reading process by an English-speaking person again.

Also, some careful proof-reading should focus on making all sentences simpler, clearer and more specific. I have given several suggestions in the minor comments, but this is a general comment that should be taken into account more broadly.

**1  Major comments**

**1.1  Clarify section 2.6 on the sub-cloud evaporation model and use it in the discussion, or remove**

- Section 2.6 needs to be clarified. For example, explain the physical basis, key equations and necessary outputs before explaining the technical details of how you calculate the inputs.

    - First explain the objectives of this model. What are the inputs? What are the outputs?
    - Then explain the physical basis for this model. Include here the key equations and explain the simplifying assumptions behind these equations.
    - Then explain the assumptions underlying how you calculate the inputs of the models. Be sure you make the difference between the hypothesis underlying the physical model, and those underlying the calculation of the inputs.

- How $\delta_0$ and $\delta_A$ are calculated needs to be explained in the text. The underlying assumption of this calculation need to be clarified.

- This model is not really used in the discussion. There are just a few lines about it l 419-423. It is not explained how the model is used, what is the purpose, how we conclude anything from it. The reader is just sent to Table 2, without any explanation of it. The model should be used in a more convincing way, otherwise, simply remove it.

**1.2  Physical meaning of $\delta_{initial}$, $\delta_{med}$ and $\Delta\delta$?**

Section 2.7

- $\Delta\delta$: can the max and min be anytime in the event? If so, what physical meaning does it have? In addition, how sensitive is it to the duration of the sample collection? e.g. longer duration for sample collection may artificially reduce $\Delta\delta$ ? And how sensitive is it to the threshold of rain amount that could be used as samples? e.g. if smaller samples are collected at the end of events, they may be more evaporatively enriched?

- Same for $\delta_{med}$: what physical meaning does it have? Why not simply using the precipitation-weighted $\delta$ , as in most studies? I expect that the precipitation-weighted $\delta$ of the event would be more representative of the large-scale vapor.

- It is assumed at several locations (e.g. l 212, l 351) that $\delta_{initial}$ is representative of the large-scale vapor. This is not convincing. Usually, $\delta_{initial}$ is affected by rainfall evaporation, because the first raindrops often fall with low rain rate and drier conditions, e.g. [Risi et al., 2010, Tremoy et al., 2014].

I advice to use precipitation-weighted $\delta$ for analysis at the inter-event scale. At the intra-event scale, clarify what $\Delta\delta$ mans or use something more physically relevant.

**1.3 Description of the results is too lengthy**

Section 3.3 is very painful to read. It would help so much the reader to present the results in a more synthetic way. The most interesting part is in the discussion, but when the reader arrives at the discussion, the results section was so long that everything is forgotten. In the results, focus on what is useful to remember to follow the subsequent discussion.

**2 Minor comments**

- l 25-27: Reword as: "During summer, the $\delta_{initial}$ values were lower dues to higher rainfall along trajectories from the Amazon forest, whereas during automn and spring, the $\delta_{initial}$ values were higher due to lower amount of rainfall along trajectories from the Atlantic Ocean and Southern Brazil."

- l 32: "meteorological" -> "isotopic"?

- l 32: "modelling" -> "model evaluation"

- l 45: "quick condensation "and formation of precipitation with substantial droplets heavy rainfall" -> "large condensation and precipitation rates" (it's more quantitative, and "substantial droplets" doesn't mean anything

- l 53-54: merge paragraph

- l 55: de Vries et al 2022 is for squall lines, so it is a convective systems. Other precipitating events have been well studies as well: e.g. mid-latitude cyclones, fronts... e.g.[Barras and Simmonds, 2009, Celle-Jeanton et al., 2004, Aemisegger et al., 2015, Thurnherr and Aemisegger, 2022, Landais et al., 2023, Muller et al., 2015]. They deserve to be cited.

- l 66: remove "and local evaporation effects", because it is not a weather system.

- l 67-69: this mixes too many different things. Reword as "High-resolution isotope information can provide a better insight into the isotopic variability during the life cycle of rainfall events".

- l 158: "Preliminary assessment of local processes" -> "Quantifying the impact of post-condensational processes". It's more specific.

- l 159-164: "Below ... conclusions.": avoid repetitions: suggestion: "Below-cloud atmospheric conditions are known to affect the rainfall composition through rain-vapor interactions. Since the isotopic composition of near-ground water vapor during the rainfall events was not measured, the framework proposed by Graf et al 2019 cannot be applied here." And then go on explaining what you do instead.

- l 217: suggested outline:
  - 3.1. Inter-event variability of meteorological and isotopic parameters
  - 3.1.1. Seasonal-mean climatic conditions
  - 3.1.2. Isotopic variations
  - 3.1.3. Moisture origin
  - 3.2 Intra-event variability of meteorological and isotopic parameters
  - 3.2.1. During summer
  - 3.2.2. During automn and spring

- l 233-234: reword as: "thermal convection over land lead to convective rainfall"

- l 279-280: "This study... lack of pattern": I don't understand this sentence. Remove. The second sentence is more specific.

- l 283: "intra-events" -> "intra-event variation of XX".
  Intra-event is an adjective, not a noun. This applies everywhere.

- l 282: "unique temporal patterns": be more specific or remove.

- l 287: "$\Delta\delta$ values for d-excess" -> "$\Delta d$" and define this earlier (sec 2.7).

- l 291: "specific local factors" -> "sub-cloud evaporation". It's more specific.

- l 293: "consistent" -> "similar"

- l 294: ". In contrast, these events showed" -> "But different d-excess evolution."

- l 304: "dis displayed in a vertical profile, illustrating these changes, with, " -> "shows"

- l 308: "parameter" -> "parameters"

- l 322: "increase trend" -> "increasing trend"

- l 346: "Detailed" -> "description"

- l 346: "were provided by both inter- and intra-events" -> "was provided at both inter- and intra-event scale"

- l 346-348: "Such... rainfall." Remove, I don't understand what it means.

- l 353: "of moist" -> "from moist"

- l 373: "representing" -> "during"

- l 375: "enhanced ... processes" -> simply "enhanced evapotranspiration"

- l 380: "Now ... its is possible ..." -> "In the extreme case where all the water vapor that is lifted by convection and condenses comes from evapotranspiration, it is possible ..."

- l 381: the assumption of isotopic equilibrium may be relevant for the first condensate, but the first condensate is not relevant to represent convective precipitation, which integrates condensation at all altitudes. This is why the calculated values are completly unrealistic for precipitation.

- l 379-390: I would replace all this calculation with unrealistic assumptions and unrealistic results by simply citing previous studies that have properly investigated the impact of evapotranspiration on the vapor and rainfall composition, e.g. [Salati et al., 1979, Worden et al., 2007, Brown et al., 2008, Levin et al., 2009, Risi et al., 2013, Worden et al., 2021]

- l 400: "Rayleigh distillation governs the depletion": clarify what this means. Depletion relative to what? What is different for this event relative to other events?

- l 402: "exchange": between what and what? Rain-vapor?

- l 405: "varying profiles": of what?

- l 407: "during a specific time of the event": be specific: which time?

- l 407: remove "was"

- l 408: "during" -> "at"

- l 413: "diverse": be more specific

- l 414: "under low humidity conditions": this sentence suggests that rainfall patterns depend mainly on RH. This is not true. Same l 421: the RH is not the only/main control on the vertical structure of rainfall. Reword to explain that both the vertical structure of rainfall and the humidity impact the local isotopic composition of rain?

- l 413-415: Reword: I think the point to make here is that when the rain evaporates in a dry environment, rain evaporation favors enrichment of the rain.

- l 435-436: "demonstrating...": Remove. Grammar problem, and not really true (convection and evapotranspiration may impact the isotopic composition even if these two processes don't interact)

- l 437: "During ... rainfall" -> "Within convective events"

- l 437: grammar problem.

- l 438-439: "The critical ... rainfall": remove or be more specific. Generally, this study doesn't convincingly argues for the impact of the vertical structure.

- l 439: "certain specific conditions of low humidity of ambient." -> "low ambient humidity."

- l 443-444: remove. This study did not investigate the conditions of convective rainfall, rather its isotopic composition.

- l 445-447: clarify. You mean that applying linear regressions based on present-day observations for paleoclimate applications should be taken with caution? Is this due to an issue with the time scale? If so reword and clarify.

**References**

[Aemisegger et al., 2015] Aemisegger, F., Spiegel, J., Pfahl, S., Sodemann, H., Eugster, W., and Wernli, H. (2015). Isotope meteorology of cold front passages: A case study combining observations and modeling. *Geophysical Research Letters*, 42(13):5652–5660.

[Barras and Simmonds, 2009] Barras, V. and Simmonds, I. (2009). Observation and modelling of stable water isotopes as diagnostics of rainfall dynamics over southeastern Australia. *J. Geophys. Res., accepted*, 114:D23308, doi:10.1029/2009JD012132.

[Brown et al., 2008] Brown, D., Worden, J., and Noone, D. (2008). Comparison of atmospheric hydrology over convective contnental regions using water vapor isotope measurements from space. *J. Geophys. Res.*, 113.

[Celle-Jeanton et al., 2004] Celle-Jeanton, H., Gonfiantini, R., Travia, Y., and Solc, B. (2004). Oxygen-18 variations of rainwater during precipitation: application of the Rayleigh model to selected rainfalls in Southern France. *J. Hydrol.*, 289:165–177.

[Landais et al., 2023] Landais, A., Agosta, C., Vimeux, F., Magand, O., Solis, C., Cauquoin, A., Dutrievoz, N., Risi, C., Leroy-Dos Santos, C., Fourré, E., et al. (2023). Abrupt excursion in water vapor isotopic variability during cold fronts at the pointe benedicte observatory in amsterdam island. *EGUsphere*, 2023:1–33.

[Levin et al., 2009] Levin, N. E., Zipser, E. J., , and Cerling, T. E. (2009). Isotopic composition of waters from Ethiopia and Kenya:Insights into moisture sources for eastern Africa. *J. Geophys. Res.*, 114:D23306, doi:10.1029/2009JD012166.

[Muller et al., 2015] Muller, C. L., Baker, A., Fairchild, I. J., Kidd, C., and Boomer, I. (2015). Intra-event trends in stable isotopes: Exploring midlatitude precipitation using a vertically pointing micro rain radar. *Journal of Hydrometeorology*, 16(1):194–213.

[Risi et al., 2010] Risi, C., Bony, S., Vimeux, F., Chong, M., and Descroix, L. (2010). Evolution of the water stable isotopic composition of the rain sampled along Sahelian squall lines. *Quart. J. Roy. Meteor. Soc.*, 136 (S1):227 – 242, DOI: https://doi.org/10.1002/qj.485.

[Risi et al., 2013] Risi, C., Noone, D., Frankenberg, C., and Worden, J. (2013). Role of continental recycling in intraseasonal variations of continental moisture as deduced from model simulations and water vapor isotopic measurements. *Water Resour. Res.*, 49:4136–4156, doi: 10.1002/wrcr.20312.

[Salati et al., 1979] Salati, E., Dall'Olio, A., Matsui, E., and Gat, J. (1979). Recycling of water in the Amazon basin: An isotopic study. *Water Resources Research*, 15:1250–1258.

[Thurnherr and Aemisegger, 2022] Thurnherr, I. and Aemisegger, F. (2022). Disentangling the impact of air-sea interaction and boundary layer cloud formation on stable water isotope signals in the warm sector of a southern ocean cyclone. *Atmospheric Chemistry and Physics Discussions*, pages 1–31.

[Tremoy et al., 2014] Tremoy, G., Vimeux, F., Soumana, S., Souley, I., Risi, C., Cattani, O., Favreau, G., and Oi, M. (2014). Clustering mesoscale convective systems with laser-based water vapor delta18O monitoring in Niamey (Niger). *J. Geophys. Res.*, 119(9):5079–5103, DOI: 10.1002/2013JD020968.

[Worden et al., 2007] Worden, J., Noone, D., and Bowman, K. (2007). Importance of rain evaporation and continental convection in the tropical water cycle. *Nature*, 445:528–532, DOI; https://doi.org/10.1038/nature05508.

[Worden et al., 2021] Worden, S., Fu, R., Chakraborty, S., Liu, J., and Worden, J. (2021). Where does moisture come from over the congo basin? *Journal of Geophysical Research: Biogeosciences*, 126(8):e2020JG006024.

---

## Author Response (AR2)

**RESPONSE LETTER**

Prof. Dr. Didier Gastmans

Environmental Studies Center – São Paulo State University

Av. 24A, 1515 – Rio Claro (SP) – Brazil

Rio Claro, November 23, 2023.

Dear. Dr. Thijs Heus,

Detailed point-by-point responses to all referee comments and specify all changes in the revised manuscript are presented below. These referee comments were essential to improve the original manuscript and have contributed for the essential modifications made in the manuscript revised version.

**Specific response to referee report 1**

**Comments in abstract:** "Combining..." is a very vague sentence. Prefer sentences that convey information on what these processes are and how they play. Remove "While". "Our results..." this sentence is a generalization that sounds a bit presumptuous. All the drivers that are considered in this study have already been explored in previous studies in other regions. I would remove.

**Response:** The abstract was completely modified. Please refer to lines 20-32.

**Comment:** Kurita 2013 used in-situ observations, not satellite observations.

**Response:** This sentence was removed because this part of introduction was re-written. Please refer to lines 42-62.

**Comment:** is this paragraph really necessary? The diel cycle is not a focus of the study anymore. This looks like a remnant of the previous version.

**Response:** The paragraph was removed. Please refer to lines 42-62.

**Comment:** section 2: according to the guidelines, all the links can be moved to a section called "Data availability" just before the Acknowledgments.

**Response:** The Data availability was included. Please refer to line 451.

**Comment:** Fig 3 is not really used in the paper. Remove the figure and this sentence?

**Response:** This figure was removed, and sentence modified. Please refer to lines 216-217.

**Comment:** grammar issue: The -> This?; try "despite no relationship being observed"? "a change in" -> "of"? Cut sentence: "This illustrates..."; subject missing.; verb missing; try "different from those observed"; sentence is too long. Shorten, cut.; "intra events" -> "events"; "control of" -> "on"; along the seasons -> depending on the season.

**Response:** All grammatical issues have been reviewed and corrected by a American English native speaker.

**Comment:** unclear sentence: try "there was similar duration, temporal $\delta^{18}O$ evolution and rain rates"; this sentence is too long.

**Response:** These sentences were rewritten. Please refer to lines 291-298.

**Comment:** General comment on the description of the figures: this is very difficult and long to read. The description should be made much shorter. You don't need to give all the values, because we can see it on the Figures. I advice to really focus on what is the main point that you are trying to make. Identify the main results and interpret it.

**Response:** The description of the figures were completely modified.

**Comment:** does this correspond to the stratiform zone of the convective system?

**Response:** Not, because there was no melting layer detected on the radar image.

**Comment:** simplify sentence, e.g. "RR and BT did not..."

**Response:** The sentence was modified. There are two new paragraphs of respectively events. Please refer to lines 328-342.

**Comment:** Fig 6 is called before Fig 5?

**Response:** Thanks for your observation. The reference to this figure has been corrected.

**Comment:** Vague and useless sentence, remove.

**Response:** It was removed.

**Comment:** Section 3.3: This is very long and messy. Paragraph l 256-261 and 288-291 could be merged. At the end of l 270 or l 287, we wonder what is the consequence on isotopes. l 298-302 need to be demonstrated. To make this section easier to read, I advice to make one paragraph per season. And for each season, use back-trajectories and meteorological conditions to interpret isotopic variations. The interpretation of isotopic variations should directly follow the description.

**Response:** To make this section easier to read, we have been modified and subdivided into results (section 3.2, please refer to line 243) and discussion sections (section 4.1, please refer to line 355).

**Comment:** clarify that this is an extreme case. If we assumed that all the vapor comes from precipitation at each recycling step, then after n recycling step, we would have $R_v = \alpha_q^n \cdot R_{v0}$, which is unrealistic.

**Response:** Yes, this is extreme case. It is evident from the preceding text in this section that the first condensate rainfall discussed in the text is produced entirely from the transpired moisture. However, we are not discussing any recycling here. The main purpose of the discussed case was to demonstrate the potential of transpired moisture in producing rainfalls characterized by elevated $\delta^2H$ and $\delta^{18}O$ values and enhanced $d$-excess values observed at Rio Claro during autumn and spring season (Fig.2).

**Comment:** this assumes that the sub-cloud layer is well-mixed. You can probably cite a paper to justify this assumption.

**Response:** References were added (Risi et al., 2019; Sarkar et al., 2023). See lines 167-168.

**Comment:** why not simply using the kinetic fractionation from Stewart 1975: $\alpha_K = (D/D_{iso})^n$

**Response:** The expression for kinetic fractionation coefficient (eq.5 in the main text) is a direct consequence of the generally accepted conceptual framework for isotope effects accompanying evaporation of water into a humid atmosphere (for details see e.g. Gat et al., 2001; Horita et al., 2008). The dependence of $\varepsilon_{kin}$ on the humidity deficit cannot be simply omitted. For $h_N = 1$ (vapour saturated atmosphere) $\varepsilon_{kin}$ vanishes, as it should be ($\alpha_{kin} = 1$).

**Comment:** isn't 1mm a bit large for a raindrop? Are there any previous studies showing disdrometer observations in this region, that could help justify this value?

**Response:** References were added (Zawadzki and Antonio, 1988; Cecchini et al., 2014). See line 196.

**Comment:** "cloud level" -> "cloud base"

**Response:** It was removed.

**Comment:** General section 3.4: what value was used for $\delta_A$? How was it chosen? How were the 2 events selected? Because they show extreme conditions? Or to contrast day and night? Clarify.

**Response:** All relevant details are reported in Table 2. Isotopic composition of water vapor below the cloud base level was calculated from the measured isotopic composition of rainfall, assuming isotopic equilibrium at (interpolated) mean temperature of local atmosphere below the cloud base:

$$\delta_A = (\delta_{rain} + 1000) * (1/\alpha_{eq}) - 1000$$

The preliminary assessment of evaporation processes were calculated for all events, now. See table 2.

**Comment:** think this was already known, already before this study. I don't think it's fair to say that isotopes variations allowed to demonstrate this. Rather, isotope variations are consistent with atmospheric dynamics in this region that have already been known for a long time. Previous studies describing the atmospheric dynamics and moisture origin for this region can be cited. e.g. for moisture origin: [van der Ent et al., 2010, Gimeno et al., 2012, Zemp et al., 2014].

**Response:** The previous conclusions have been revised. The current focus is on how regional moisture transport controls the initial isotopic composition. This study differs from previous studies mentioned in that there has been no use of high-frequency sampling.

**Comment:** "It's hard to overestimate": strange and vague sentence. Rather, cite previous studies on the impact of deforestation to make a more precise statement.

**Response:** The sentence was removed.

**Comment:** "reduction in precipitation amount": there is an extensive literature on the impact of deforestation on South American rainfall, and results are not always obvious and consistent. These previous studies should be cited to make a more precise statement.

**Response:** The sentence was removed.

**Comment:** "high-resolution": it's not clear to me what was the added value of the high-resolution sampling. What conclusion couldn't have been drawn if you had done only event-scale sampling? This needs to be clarified.

**Response:** To clarify this, we aim to examine the disparities between the initial, median and range of isotopic values ($\Delta\delta$) of intra-events. Therefore, the interpretation was divided into inter and intra-event analysis to maximize the level of detail and comprehension of the changes from regional processes to during the rain event.

**Comment:** This fig is too small, it's hard to read. Maybe cut it into 2 pieces?

**Response:** Yes, this figure was modified in two pieces. See figures 5 and 6.

**Comment:** why discussing only the impact on *d*-excess, an not also on deltas?

**Response:** The discussion of delta was included throughout section 4.2. The semi-quantitative assessment we used focused on *d*-excess due to the greater range of values in relation to the deltas, and to avoid repetitive explanations.

**Specific response to referee report 2**

**General comments:** As suggested by a reviewer, an English speaker reviewed the manuscript, and it was reorganized. The method section now includes the semi-quantitative assessment, and the results are

divided into sections for inter-event and intra-event data. The discussion is separated into regional and local processes, which detail the main controls on isotopic variability. We maximized the level of detail in our sampling strategy, gaining an understanding of the regional processes and changes during rainfall events. For this, we was examined the contrasts among the initial, median, and range of isotopic values ($\Delta\delta$) within events. Upon recommendation by the reviewers, the diurnal variations were omitted from the analysis.

**Comment:** Abstract needs to be rewritten.
**Response:** Abstract was entirely rewritten, and all suggestions were accepted.

**Comment:** Word not a proper word used here.
**Response:** The word was removed.

**Comment:** "climat projections" is too general. These projections are model simulations or actual observation and so on. Here, you should be very specific about this
**Response:** The model was pre-CMIP6 and was included in the text. Please refer to line 39.

**Comment:** This sentence is too long. For the convenience of readers, I would split it into two sentences.
**Response:** This sentence was modified. Please refer to lines 42-50.

**Comment:** In Southeast Asia region, convective events occurring in the afternoon are largely local, and of course large-scale convection does occur in th afternoon. For regional convectie events like Squalls normally occur in the early morning. Please see this paper: He, S., Goodkin, N. F., Kurita, N., Wang, X., & Rubin, C. M. (2018). Stable isotopes of precipitation during tropical Sumatra Squalls in Singapore. Journal of Geophysical Research: Atmospheres, 123, 3812–3829. https://doi.org/10.1002/ 2017JD027829. I don't think your sampling strategy can answer this diurnal effect on isotopes.
**Response:** As the daytime differences has been omitted, we have not included the corresponding references.

**Comment:** There are more papers talking about squall lines in tropics apparently you missed it.
**Response:** Yes, thanks for suggestions. However, we've only added a few examples so as not to pollute the text too much.

**Comment:** These papers are not in the reference list at the end.
**Response:** Risi et al., 2010 and Tremoy et al., 2014 were included in reference list.

**Comment:** "belongs to" is not an appropriate word. GNIP was initially established by IAEA and WMO with participation of many member states. You should go to IAEA website to chech the background of GNIP.

**Response:** The sentence was rewritten. Please refer to lines 77-78.

**Comment:** rephrase the sentence to make it more clear

**Response:** The sentence was rewritten. Please refer to lines 88-89.

**Comment:** This sentence sounds like that you collected day-time and night-time samples at different locations. Please rephrase this sentence to make it clear.

**Response:** The sentence was rewritten. Please refer to lines 89-90.

**Comment:** I think 10 days back is a little bit too long. What is the average retention time of moisture in the air at your region?

**Response:** Around 10 days. The articles cited in the sentence justify the use of 10 days (Gimeno et al., 2010, 2020; van der Ent and Tuinenburg, 2017).

**Comment:** I didn't find this paper in the reference list.

**Response:** dos Santos et al., 2023 was included.

**Comment:** Did you also investigate monthly precipitation and analyze its isotopes during your study?

**Response:** Yes. However, we solely utilize the monthly isotopic composition of precipitation depicted in Figure 2 to characterize the seasonal pattern. The analytical information was identical to that of high-frequency samples. The details regarding the monthly samples and their analysis have been incorporated into the explanation of Figure 2.

**Comment:** Please try to organize this section discussion in more systematic way for readers to follow.

**Response:** Section 3.2 was divided into two parts: results (section 3.3, which separated the season events) and discussion (section 4.2).

**Comment:** This sentence has some grammar issues and please rephrase it. Also, your description of trends is not that accurate. For example, increase trend meand that value from low at the begining to higher at the end, but it is not the case you described. Likewise, decrease trend mean values from higher to lower from start to end. It is more appropriate to say that the isotope values varied through these event. Anyway, I am confused with what you are talking about in sentence.

**Response:** We completely agree. The sentence has been rewritten. Please refer to lines 286-289**.**

**Comment:** so you put events from different years in the same group, although they both were from Feb? I dont know about this statemet, as you compared the event from different years, not the same day at the same year over 24 hour. If this is true, then how you can explain the event 2020/201 and 2020/01/20. Did you check the data from other seasons, I mean the difference between day and night in other seasons?

**Response:** We have organized the events by season to enhance readability. Nevertheless, in this updated version, we display the findings of all intra-events, elucidating each one, irrespective of the sampling period (day, night, or month).

**Comment:** It is no that obvious though and both exhibit no change

**Response:** The variation of these events has been explained in detail. Please refer to lines 322-323.

**Comment:** Change in 4v is clear but not in 4u.

**Response:** The variation of the event has been explained in detail. Please refer to lines 330-332.

**Comment:** This section is kind of wordy. Things discussed are largely known already and not new to readers anymore.

**Response:** Section 3.3 was divided into two parts: results (section 3.2) and discussion (section 4.1). Consequently, there was rewritten.

**Comment:** We can obtain the similar conclusions using daily or monthly precipitation isotope data. Do need to have high-resolution event isotope data.

**Response:** Agree. Therefore, we connected the local and regional processes by utilizing the disparity between the initial, median, and range of isotopic values. The intra-event analysis illustrated the impact of cloud evolution during events and how these local mechanisms altered the regional isotopic composition. This change can only be discerned through intra-event data collection, which is why a significant portion of the article concentrates on this aspect.

**Comment:** This part needs to reorganized. The detailed methods and calculations should be located to method section and here, you should focus on the results and your hypothesis. You should look at all the events examined in this study. The arguments in this section are also very weak. Also, the water line defined by water isotopes of earch event can tell the rain evaporation.

**Response:** The methodology and computations are now included in section 2.6 of the methods chapter, which has been updated and embellished. Calculations have been made for all events studied. The arguments are justified in respected articles within the field of isotopy. It is solely from the knowledge acquired through these articles that we recommend determining the sub-cloud process. This semi-quantitative determination is not used as a primary means of understanding the change in local isotopic

composition due to a lack of data. We are only using it as a secondary evaluation to support the interpretation that is derived from the intra-event isotopic evaluation. Despite this, we have included this section in the text as it is essential to enhance the discussion and quantify the local isotopic processes studied. We opted not to use the local meteoric water line for each event, due to limited isotopic value variation in some cases. Consequently, the points are closely grouped, resulting in a meaningless straight line.

**Comment:** The first sentence is too long and please try to split into two.
**Response:** The sentence was rewritten. Please refer to lines 161-162.

**Comment:** This sentence is redundant as it is not informative at all.
**Response:** The sentence was removed.

**Comment:** Did you check the correlation between rainfall amount and rain isotopes?
**Response:** Yes, the correlations were weak (r < 0.5).

**Comment:** some papers cited in the main text are not in the reference list!
**Response:** All papers cited were checked in the reference list.

**Comment:** The style of paper's titles should be consistent; the sentence style is common but uppercase for all words is not very common.
**Response:** The style of paper's titles was corrected.

**Comment:** Table 2: Why you only look at two events?
**Response:** Now, we look all events. Please refer to Table 2.

**Comment:** The figure caption can be better.  Please rephrase it.
**Response:** The figure caption was modified. Please refer to Figure 1.

**Comment:** Figure 2: The figure caption can be better.  Please rephrase it. Are these monthly precipitation isotope data from your own observation or obtained from others?  Please indicate this infomation in the caption or somewhere in the main text. Please indicate the sources of these data or if not, how you calculated them.
**Response:** The information was included. Please refer to Figure 2 (line 755).

**Comment:** Figure 4: the marks on y-axis are too crowded, and it is hard to read them.
**Response:** this figure was modified in two pieces. See figures 5 and 6.

**Comment:** some sub-figures have two arrows in opposite direction and can you explain more?

**Response:** The arrows were removed.

---

## Author Response (AR3)

Prof. Dr. Didier Gastmans

Environmental Studies Center – São Paulo State University

Av. 24A, 1515 – Rio Claro (SP) – Brazil

Rio Claro, February 27, 2024.

Dear. Dr. Thijs Heus,

We appreciate all the extensive work on the several revisions provided to improve our manuscript. In this version, we have attended all the minor and major comments provided by a single reviewer. Two faculty members at US-based institutions have reviewed the final English style. The main change in the article relates to the intra-event evaluation, where the results and figures in this section have all been reworked and improved. The figures speak for themselves, and only the main message is presented in the text, avoiding the excessive use of data-rich paragraphs. Sentences were shortened to make this reading more enjoyable. The results section was reduced by at least two pages, aiming for a more concise scientific average. Detailed point-by-point responses to the single referee's comments are presented below.

**Major comments**

1.1 Clarify section 2.6 on the sub-cloud evaporation model and use it in the discussion, or remove.

**Response:** As suggested by the reviewer the entire model section was removed.

1.2 Physical meaning of $\delta_{initial}$, $\delta_{med}$ and $\Delta\delta$? Section 2.7

$\Delta\delta$: can the max and min be anytime in the event? If so, what physical meaning does it have? In addition, how sensitive is it to the duration of the sample collection? e.g. longer duration for sample collection may artificially reduce $\Delta\delta$? And how sensitive is it to the threshold of rain amount that could be used as samples? e.g. if smaller samples are collected at the end of events, they may be more evaporatively enriched?

**Response:** Since the model section was removed, $\Delta\delta$ is not present in the text anymore.

Same for $\delta$med: what physical meaning does it have? Why not simply using the precipitation-weighted $\delta$, as in most studies? I expect that the precipitation-weighted $\delta$ of the event would be more representative of the large-scale vapor.

**Response:** The weighted average is most used in studies with monthly and daily data. However, we understand your point. Since we were doing an inter-event assessment, the weighted average represents the weight of rainfall, so the processes that become more apparent are ultimately regional. In this context, we have included the weighted average to discuss regional processes, accepting the reviewer's suggestion.

It is assumed at several locations (e.g. l 212, l 351) that $\delta$initial is representative of the large scale vapor. This is not convincing. Usually, $\delta$initial is affected by rainfall evaporation, because the first raindrops often fall with low rain rate and drier conditions, e.g. [Risi et al., 2010, Tremoy et al., 2014].

**Response:** While potential initial evaporation has been discussed in the literature, based on the relatively high RH values over the study area and in the vertical profiles, its effects are most likely minimal. Also, many studies forget that rain does not form under the collection point, so the idea that the evaporation process influences the isotopic composition is more appropriate for this initial condition, the first rain after the formation of the system. In our case, and many others, the rain has already formed and is passing over the collection point. A few kilometers before the rain system arrives, the humidity and temperature begin to change, increasing and decreasing as the rain arrives. The question is: how much of this change in humidity is enough to cause evaporation in a growing or mature convective rain system? I think this would be very difficult to answer without vapor isotope data. In this context, we believe that the initial value represents the mix of regional processes before the arrival of this system at the collection point. This system mixes with local conditions as it passes over the sampling point, resulting in the observed isotopic variation pattern. To better clarify these comments, a specific study would be more appropriate, so we have decided to also remove the $\delta_{initial}$ parameter from this version of the article and focus on how local controls produce changes in the isotopic variation pattern of rain.

I advise to use precipitation-weighted $\delta$ for analysis at the inter-event scale. At the intra-event scale, clarify what $\Delta\delta$ mans or use something more physically relevant.

**Response:** Recommendation accepted, excluding all isotopic parameters ($\delta_{initial}$ and $\Delta\delta$) from the previous version and using the weighted average as the main parameter for inter-event analysis.

1.3 Description of the results is too lengthy

Section 3.3 is very painful to read. It would help so much the reader to present the results in a more synthetic way. The most interesting part is in the discussion, but when the reader arrives at the discussion, the results section was so long that everything is forgotten. In the results, focus on what is useful to remember to follow the subsequent discussion.

**Response:** We recognize how painful it was to read this section with data-rich paragraphs. The overall section was smoothed and reduced by almost 50%. The reading is more enjoyable now.

**2 Minor comments**

Abstract

Reword as: During summer, the $\delta$initial values were lower dues to higher rainfall along trajectories from the Amazon forest, whereas during automn and spring, the $\delta$initial values were higher due to lower amount of rainfall along trajectories from the Atlantic Ocean and Southern Brazil.

**Response:** This phrase was removed.

Meteorological > isotopic?

**Response:** The word was modified.

Modelling > model evaluation

**Response:** The word was modified.

Introduction

"quick condensation and formation of precipitation with substantial droplets heavy rainfall" - > "large condensation and precipitation rates" (it's more quantitative, and substantial droplets doesn't mean anything.
**Response:** The sentence was modified.

merge paragraph (weather systems)
**Response:** The paragraphs were merged.

de Vries et al 2022 is for squall lines, so it is a convective systems. Other precipitating events have been well studies as well: e.g. mid-latitude cyclones, fronts... e.g.[Barras and Simmonds, 2009, Celle-Jeanton et al., 2004, Aemisegger et al., 2015, Thurnherr and Aemisegger, 2022, Landais et al., 2023, Muller et al., 2015]. They deserve to be cited.
**Response:** The cited articles were not included in the section recommended by the reviewer, which focuses on convective processes and their impact on isotopic composition. Instead, they were added to the subsequent paragraphs that mentioned atmospheric systems studied at high frequency.

remove "and local evaporation effects", because it is not a weather system
**Response:** The sentence was modified.

this mixes too many different things. Reword as "High-resolution isotope information can provide a better insight into the isotopic variability during the life cycle of rainfall events".
**Response:** The sentence was modified.

Data and methods

"Preliminary assessment of local processes" -> "Quantifying the impact of post-condensational Processes". It's more specific. "Below ... conclusions.": avoid repetitions: suggestion: "Below-cloud atmospheric conditions are known to affect the rainfall composition through rain-vapor interactions. Since the isotopic composition of near-ground water vapor during the rainfall events was not measured, the framework proposed by Graf et al 2019 cannot be applied here." And then go on explaining what you do instead.
**Response:** The section 2.6 was deleted.

Results: The outline suggested by the reviewer was accepted, and the entire section on intra-event results has been rewritten.

Reword as: "thermal convection over land lead to convective rainfall"
**Response:** The sentence was modified.

Discussion

"Detailed" > "description"
**Response:** The word was modified.

"were provided by both inter- and intra-events" > "was provided at both inter- and intra-event Scale"

**Response:** The sentence was modified.

"were provided by both inter- and intra-events" -> "was provided at both inter- and intra-event scale

**Response:** The sentence was modified.

"Such... rainfall". Remove, I don't understand what it means.

**Response:** The sentence was deleted.

"of moist" -> "from moist"

**Response:** The word was modified.

"representing" > "during"

**Response:** The word was modified.

Discussion – regional atmospheric controls

"enhanced… processes" > simply "enhanced evapotranspiration"

**Response:** The sentence was modified.

"Now ... its is possible ..." -> "In the extreme case where all the water vapor that is lifted by convection and condenses comes from evapotranspiration, it is possible ..."

**Response:** The sentence was deleted because equation 7 was deleted, too.

About equation 7: "the assumption of isotopic equilibrium may be relevant for the first condensate, but the first condensate is not relevant to represent convective precipitation, which integrates condensation at all altitudes. This is why the calculated values are completely unrealistic for precipitation".

"I would replace all this calculation with unrealistic assumptions and unrealistic results by simply citing previous studies that have properly investigated the impact of evapotranspiration on the vapor and rainfall composition, e.g. [Salati et al., 1979, Worden et al., 2007, Brown et al., 2008, Levin et al., 2009, Risi et al., 2013, Worden et al., 2021]".

**Response:** We completely agree with the reviewer. Equation 7 has been removed. The entire explanation has been modified, and all the suggested references have been included.

Discussion – Local atmospheric controls

One of the reviewer's main questions was: "Reword to explain that both the vertical structure of rainfall and the humidity impact the local isotopic composition of rain?"

**Response:** To explain how the vertical profile of the rain influences the isotopic composition, we decided further to investigate the radar reflectivity (Z) data. To do this, we quantified the Z values in the vertical profile as described in Methods section 2.5. By quantifying the variation in Z, we can see that changes in its values illustrate the process of capturing water particles in the raindrop, so that the more it is captured, the higher the concentration

of water in the raindrop and, consequently, the higher the reflectivity. Events with a vertical variation in Z indicate a change in this droplet formation process. We can see that these changes correspond to large variations in the $\delta^{18}O$ values, especially in the d-excess values. These variations are confirmed by the significant correlations found between Z, $\delta^{18}O$, and $d$-excess. Regarding the influence of humidity, we believe that more data from the vertical humidity profile measured in situ could help, as, unfortunately, it is not possible to have them; we believe that the influence of humidity on intra-events ends up being minimal because even at d-excess values of less than 10, the humidity values at the surface were above 90%. For more details on the local meteorological controls, see section 4.2 and Figures 5-8.

Conclusion

"demonstrating..": Remove. Grammar problem, and not really true (convection and evapotranspiration may impact the isotopic composition even if these two processes don't interact)

**Response:** The sentence was modified.

"During ... rainfall"-> "Within convective events"; "grammar problem"

**Response:** The sentence was modified.

"The critical ... rainfall": remove or be more specific. Generally, this study doesn't convincingly argues for the impact of the vertical structure".

**Response:** The sentence was modified, and the impact of the vertical structure was included.

"certain specific conditions of low humidity of ambient." -> "low ambient humidity."

**Response:** The sentence was removed.

"remove. This study did not investigate the conditions of convective rainfall, rather its isotopic composition".

**Response:** The sentence was removed.

"clarify. You mean that applying linear regressions based on present-day observations for paleoclimate applications should be taken with caution? Is this due to an issue with the time scale? If so reword and clarify."

**Response:** The sentence was modified.